# H4K20me3 is important for Ash1-mediated H3K36me3 and transcriptional silencing in facultative heterochromatin in a fungal pathogen

**Mareike Möller** [¤a]*, **John B. Ridenour** [¤b], **Devin F. Wright**, **Faith A. Martin**, **Michael Freitag**

Department of Biochemistry and Biophysics, Oregon State University, Corvallis, Oregon, United States of America

¤a Current address: Research School of Biology, The Australian National University, Canberra, Australia
¤b Current address: Cell Cycle and Cancer Biology Research Program, Oklahoma Medical Research Foundation, Oklahoma City, Oklahoma, United States of America
* mareike.moeller@anu.edu.au

**Data Availability Statement:** Sequencing raw reads (FASTQ files) of all ChIP-seq and RNA-seq are available online at Sequence Read Archive (SRA) under BioProject ID PRJNA902413.

## Abstract

Facultative heterochromatin controls development and differentiation in many eukaryotes. In metazoans, plants, and many filamentous fungi, facultative heterochromatin is characterized by transcriptional repression and enrichment with nucleosomes that are trimethylated at histone H3 lysine 27 (H3K27me3). While loss of H3K27me3 results in derepression of transcriptional gene silencing in many species, additional up- and downstream layers of regulation are necessary to mediate control of transcription in chromosome regions enriched with H3K27me3. Here, we investigated the effects of one histone mark on histone H4, namely H4K20me3, in the fungus *Zymoseptoria tritici*, a globally important pathogen of wheat. Deletion of *kmt5*, the gene encoding the sole methyltransferase responsible for H4K20 methylation, resulted in global derepression of transcription, especially in regions of facultative heterochromatin. Derepression in the absence of H4K20me3 not only affected known genes but also a large number of novel, previously undetected transcripts generated from regions of facultative heterochromatin on accessory chromosomes. Transcriptional activation in *kmt5* deletion strains was accompanied by a complete loss of Ash1-mediated H3K36me3 and chromatin reorganization affecting H3K27me3 and H3K4me2 distribution in regions of facultative heterochromatin. Strains with H4K20L, M or Q mutations in the single histone H4 gene of *Z. tritici* recapitulated these chromatin changes, suggesting that H4K20me3 is important for Ash1-mediated H3K36me3. The Δ*kmt5* mutants we obtained were more sensitive to genotoxic stressors than wild type and both, Δ*kmt5* and Δ*ash1*, showed greatly increased rates of accessory chromosome loss. Taken together, our results provide insights into an unsuspected mechanism involved in the assembly and maintenance of facultative heterochromatin.

Normalized bigwig files for all ChIP-seq datasets, reference genome, and annotation files have been deposited at zenodo under doi: https://zenodo.org/record/8218851.

**Funding:** MM was supported by a grant from the Deutsche Forschungsgemeinschaft (DFG, grant number MO 3755/1-1). JBR was supported by a grant from the United States Department of Agriculture-Agriculture and Food Research Initiative (USDA-AFRI, grant number 2019-67012-29722). Chromatin research in the Freitag lab is supported by grants from the National Science Foundation (NSF, grant number MCB1818006), National Institutes of Health (NIH, grant number R01GM132644), and the United States-Israel Binational Science Foundation (BSF, grant number #2019034) to MF. The funders had no role in study design, data collection and analysis, decision to publish, or preparation of the manuscript.

**Competing interests:** The authors have declared that no competing interests exist.

## Author summary

Facultative heterochromatin contains genes important for specific developmental or life cycle stages. Transcriptional regulation of these genes is influenced by chromatin structure. Here, we report that trimethylation of lysine 20 on histone H4 (H4K20me3), is enriched in facultative heterochromatin and important for transcriptional repression in these regions in an important agricultural pathogen. Furthermore, normal levels of H4K20me3 are essential for deposition of another repressive histone mark, Ash1-mediated H3K36me3, and affect the distribution of other marks including H3K27me3. We conducted a genome-wide assessment of H4K20 trimethylation levels in a fungus, and our discoveries reveal that multiple chromatin modifications are required to establish transcriptional silencing, providing the framework to understand epistasis relationships among these histone marks.

## Introduction

Chromatin, the assembly of DNA, RNA, and proteins that constitutes chromosomes, can assume active and inactive states that are correlated with different histone and DNA modifications [1]. Transcriptionally inactive, "silent" chromatin is separated into "constitutive heterochromatin" and "facultative heterochromatin". Constitutive heterochromatin is most often marked by H3K9me2/3 and DNA methylation and found within or near centromeres, subtelomeric regions or telomeric repeats, rDNA, and transposable elements. Facultative heterochromatin is typically enriched with H3K27me3 and found on specific genes often associated with development [2,3], in large broad local enrichments (BLOCs) in mice [4], or across long sections of chromosomes in many fungi [5,6]. Facultative heterochromatin can be more dynamic than constitutive heterochromatin and "on" or "off" states vary between cell types or individuals within a species [7]. While H3K27me3, mediated by Polycomb Repressive Complex 2 (PRC2), is considered to be a hallmark histone modification correlated with facultative heterochromatin [8], little is known about other chromatin marks that are important for formation, maintenance, and gene silencing in these regions. In animals, the PRC1 complex and H2AK119ub1 play an important role in PRC2 and H3K27me3 recruitment [9], but so far there is no evidence for a canonical PRC1-like complex in fungi [10].

Other histone modifications found in transcriptionally silent regions of facultative heterochromatin include H4K20 and H3K36 methylation. Depending on the organism, one or more histone methyltransferases (HMTs) are responsible for methylating H4K20 [11]. In animals, one HMT mediates H4K20me1, for example, KMT5A in humans and PR-SET-7 in *Drosophila melanogaster* [12,13], and at least one HMT mediates H4K20me2/3, for example KMT5B/C in humans and Su(var)4-20 in *D. melanogaster* [14,15]. In contrast, a single enzyme, Kmt5, called Set9 in *Schizosaccharomyces pombe* and SET-10 in *Neurospora crassa*, catalyzes all H4K20me in fungi [16–18]. The presence of different methylation states of H4K20 (i.e., me1, me2, or me3) is essential for development and associated with DNA repair, replication, cell cycle control, genome stability, and chromatin compaction in animals [11,19–24]. In contrast, there are few studies on the function of H4K20me in fungi. In *S. pombe*, H4K20me is important for DNA repair by recruitment of Crb2 to sites of DNA damage [16]. Studies with the filamentous plant pathogenic fungi *Fusarium graminearum*, *F. fujikuroi*, and *Magnaporthe oryzae* (syn. *Pyricularia oryzae*) revealed minor, if any, effects on virulence [17,18], and only subtle and species-specific effects on expression of secondary metabolite cluster genes and tolerance to stress

conditions in Δ*kmt5* strains [17]. Notably, none of the previous studies analyzed the genome-wide distribution of any of the H4K20me modification states in fungi.

In contrast, H3K36me is a widespread and well-studied histone modification deposited by Set2 in an RNAPII-dependent manner during transcription elongation [25]. In addition to Set2, a group of RNAPII-independent SET-domain proteins, the ASH1 or NSD family proteins, catalyze H3K36 methylation in many organisms [26,27]. ASH1-mediated H3K36me has been found in transcriptionally repressed regions linked to Polycomb-mediated repression and is thought to counteract PRC2-mediated H3K27me3 [28,29]. In fungi, ASH1-mediated H3K36me3 occurs in transcriptionally repressed regions enriched with H3K27me3 and is important for normal distribution of this mark, at least in *N. crassa* and *F. fujikuroi* [30–32].

In this study, we sought to address how H4K20me3 and ASH1-mediated H3K36me3 contribute to the formation, maintenance, and silencing of facultative heterochromatin. The filamentous fungus *Zymoseptoria tritici*, a global pathogen of wheat, has a well-characterized epigenetic landscape that allowed us to differentiate euchromatin, constitutive or facultative heterochromatin, regions with cytosine methylation (5mC), short regional centromeres, subtelomeric regions, and telomeric repeats [6,33,34]. Facultative heterochromatin in *Z. tritici*, defined here as chromatin enriched with H3K27me3 and transcriptionally silenced, covers subtelomeric regions and accessory chromosomes (also referred to as "non-essential" or "conditionally dispensable" chromosomes), which create variation between isolates [6,35]. While H3K27me3 in these regions is involved in chromosome stability and accumulation of mutations, loss of H3K27me3 has only a minor impact on transcriptional activation [33,36]. Therefore, we hypothesized that another repressive histone modification exists that is epistatic to H3K27me in *Z. tritici* and perhaps other organisms.

Here, we show that H4K20me3 is one such repressive modification and is generated exclusively by the SET-domain HMT, Kmt5. Kmt5 and H4K20me3 are crucial for transcriptional silencing in regions of facultative heterochromatin and important for Ash1-mediated H3K36me3, thereby contributing significantly to the formation of H3K27me3-enriched facultative heterochromatin.

## Results

### Kmt5 and Ash1 are important for normal growth but not essential in *Zymoseptoria tritici*

We identified *kmt5* (Zt_chr_3_00475) and *ash1* (Zt_chr_13_00232) in the genome of the reference isolate IPO323 by BLAST searches with *S. pombe* SET9 and *N. crassa* ASH-1 sequences as baits. We found a single H4K20 methyltransferase homolog, consistent with findings in other fungi but different from metazoans, where multiple enzymes are involved in catalyzing H4K20me1 and H4K20me2/3 (Fig 1A). Protein sequence alignments of the SET domains of H4K20 methyltransferases from *Z. tritici*, *S. pombe*, *D. melanogaster*, *Danio rerio*, and human revealed higher sequence similarity between Kmt5 in fungi and known H4K20me2/3 methyltransferases than H4K20me1 methyltransferases (S1 Fig). We showed that Kmt5 in *Z. tritici* is most likely responsible for all H4K20 methylation, as *kmt5* deletion strains are unable to generate either H4K20me1 or H4K20me3 (Figs 1B and S2).

There is a single gene encoding a *Z. tritici* Ash1 homolog. This Ash1 protein lacks, as does *Neurospora* ASH-1 [30], the C-terminal BROMO, PHD, and BAH domains found in animal Ash1 homologs but contains the conserved AWS, SET, and post-SET domains (Fig 1C). While the specific functions of the additional domains in most animal Ash1 proteins are not known, the PHD and BAH domains are essential for Ash1 to counteract Polycomb-mediated repression in *D. melanogaster* [37].

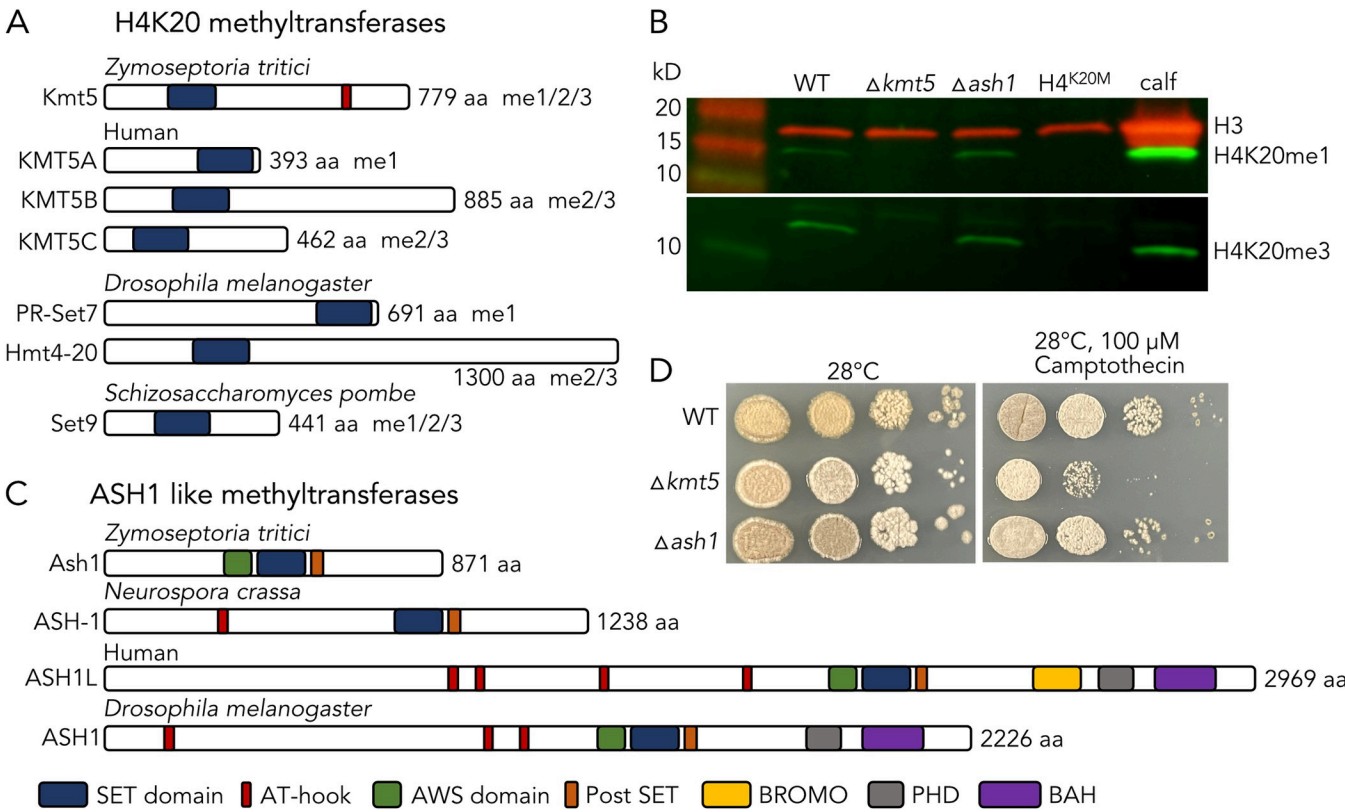

**Fig 1. Domain structure and characterization of defects related to two histone methyltransferases involved in generation of facultative heterochromatin.**
**A)** Comparison of H4K20 methyltransferases related to Kmt5 (accession numbers in S1C Fig). The blue box denotes the SET domain. **B)** Deletion of *kmt5* and substitution of H4 lysine 20 with methionine (H4K20M) causes complete loss of H4K20 methylation but *ash1* deletion has no effect on H4K20 methylation. Calf thymus histones were used as a control. **C)** Comparison of H3K36 methyltransferases related to Ash1. **D)** *kmt5* but not *ash1* mutants are mildly affected by the DNA replication inhibitor camptothecin at high temperature.

To determine the function of Kmt5 and Ash1, we deleted the respective genes in the wild-type isolate Zt09 [38] by replacement with a gene encoding hygromycin phosphotransferase (*hph*), conferring resistance to hygromycin B. Mutant strains were confirmed by PCR, Southern assays (S3 Fig), and sequencing. Both Δ*kmt5* and Δ*ash1* mutants exhibited phenotypic differences compared to the wild-type strain. The Δ*kmt5* mutant showed increased sensitivity to genotoxic stress and altered morphology, such as differences in pigmentation and hyphal growth at higher temperatures (28˚C versus the usual 18˚C; Fig 1D). Deletion of *ash1* only slightly retards growth and had overall minor effects on growth under the tested conditions (Figs 1D and S4).

To complement Δ*kmt5*, we replaced the *hph* gene with a wild-type *kmt5* gene (amplified from isolate Zt469 [34]) and *neo* resistance gene, conferring resistance to G418. To complement the Δ*ash1* strain, we replaced the *hph* resistance gene with an *ash1-gfp-V5* construct and *neo* resistance gene. To verify complementation strains restored histone methyltransferase activity, we performed ChIP-seq experiments and phenotyping assays. Complementation strains showed wild-type-like H4K20me3 and H3K36me3 enrichment and wild-type-like growth under the tested conditions (S5 Fig).

## Loss of Kmt5 or Ash1 results in increased chromosome loss

Previously, we measured accessory chromosome loss as an indicator of overall genome stability [33]. A four-week growth experiment including five replicate cultures of wild-type, Δ*kmt5*,

**Table 1. Accessory chromosome loss in wild type (WT), Δkmt5, and Δash1 strains.** Chromosome loss is tabulated for individual chromosomes and as the sum of all losses (loss), compared to strains without chromosome loss (no loss). Overall chromosome loss frequency (%) was calculated as (loss/total tested) *100. Some strains lost multiple chromosomes.

|  | WT | Δkmt5 | Δash1 |
|---|---|---|---|
| chromosome 14 | 2 | 107 | 17 |
| chromosome 15 | 3 | 4 | 1 |
| chromosome 16 | 2 | 60 | 89 |
| chromosome 17 | 0 | 4 | 0 |
| chromosome 19 | 0 | 0 | 0 |
| chromosome 20 | 1 | 0 | 2 |
| chromosome 21 | 0 | 0 | 0 |
| total tested | 240 | 240 | 240 |
| loss | 8 | 140 | 99 |
| no loss | 232 | 100 | 141 |
| loss frequency (%) | 3 | 58 | 41 |

and Δ*ash1* strains revealed an increase in accessory chromosome loss in both mutants (Table 1). Increased chromosome loss rates did not affect all accessory chromosomes equally; chromosomes 14 and 16 in Δ*kmt5*, and chromosome 16 in Δ*ash1* were lost more frequently than any other accessory chromosome. This indicates that both Kmt5 and Ash1 are important for the maintenance of at least certain accessory chromosomes. We have not found any specific characteristics that make some accessory chromosomes more prone to be lost in certain mutant backgrounds.

## ChIP-seq uncovers different heterochromatin states in *Z. tritici*

To determine the effects on chromatin in both Δ*kmt5* and Δ*ash1* mutants, we first performed ChIP-seq of selected histone modifications (H3K4me2, H3K9me3, H3K27me2, H3K27me3, H3K36me3, H4K20me3) associated with eu- and heterochromatin in a wild-type strain. We found that H4K20me3 is widespread along chromosomes, enriched across gene bodies, and exhibits pronounced enrichment in regions we previously characterized as facultative hetero-chromatin, based on low levels of gene expression and the presence of H3K27me3 [6,33,34] (Figs 2A and S6). We also tested the localization of H3K27me2 in *Z. tritici* and found that it is present in the same regions as H3K27me3 but at lower levels of enrichment. H3K36me3 showed a similar pattern and distribution as H4K20me3 and both marks overlap in some euchromatic and facultative heterochromatic regions but were absent from constitutive het-erochromatin marked by H3K9me3 alone or by a combination of H3K9me3 and H3K27me2/3 (Figs 2A and S6). To characterize modifications enriched in facultative heterochromatic regions in more detail, we plotted enrichment of H4K20me3 and H3K36me3 within 500 bp windows of regions that are enriched with H3K27me3 in wild type (Fig 2B). We found two dis-tinct clusters within H3K27me3-enriched regions: cluster 1 was enriched with H4K20me3 and H3K36me3, while cluster 2 exhibited very low levels of H3K36me3 and H4K20me3. Further analyses of the genomic context of these clusters revealed that cluster 1 mostly overlaps with genes and intergenic regions, while cluster 2 overlaps with transposable elements (TEs) that are also enriched with H3K9me3, and intergenic regions (Fig 2C). H3K27me3 and H3K9me3 co-occur in regions close to the telomeres, some repeats, and sections of accessory chromo-somes; the common feature of these regions are TEs. Both H3K36me3 and H4K20me3 are pre-dominantly excluded from these regions.

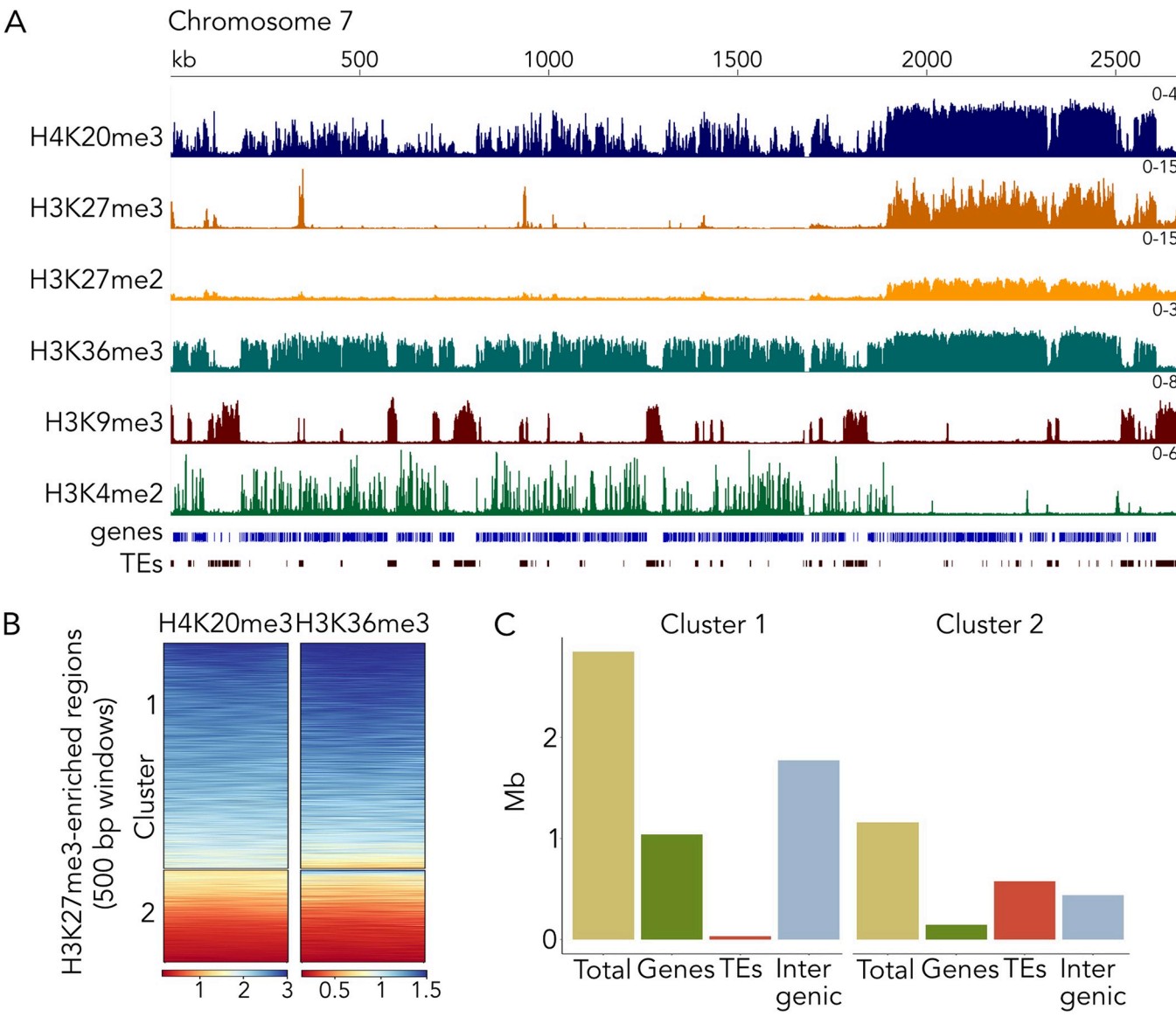

**Fig 2. Characterization of different histone modification patterns and their genomic context. A)** Distribution of selected histone modifications on chromosome 7 in the wild-type *Zymoseptoria tritici* strain. In facultative heterochromatin regions, H4K20me3, H3K27me2/3 and H3K36me3 overlap. **B)** Enrichment of H3K36me3 and H4K20me3 within H3K27me3 regions. There are two distinct clusters, regions where H3K36me3 and H4K20me3 co-occur (cluster 1) and regions where both marks are absent or only one of them is present (cluster 2). Windows in the heatmap are ordered from highest enrichment (blue) to lowest enrichment (red). **C)** Genomic context of cluster 1 and 2; Mb, million base pairs; TEs, transposable elements.

These two clusters may signify functionally different types of facultative heterochromatin, with cluster 1 representing the traditionally accepted type of facultative heterochromatin associated with conditional gene repression in some organisms, while cluster 2 represents constitutive heterochromatin doubly marked with H3K9me3 and H3K27me3 [6]. Based on these results we now differentiate at least three types of heterochromatin in *Z. tritici*: (1) constitutive heterochromatin, correlated with H3K9me3 enrichment only and enriched on repeats and TEs, (2) constitutive heterochromatin, correlated with enrichment of H3K9me3 and H3K27me3, found close to telomeres and some repeats and TEs, especially on accessory chromosomes, and (3) facultative heterochromatin, correlated with H3K27me3, H3K36me3, and

H4K20me3 enrichment, mostly found on genes and intergenic regions either in large blocks or in shorter segments dispersed on chromosomes. The functional relevance of these states remained to be determined and here we began these investigations.

## Kmt5 is important for Ash1-mediated catalysis of H3K36me3 and normal H3K27me3 distribution

Deletion of *kmt5* resulted in complete loss of H4K20me3 (Figs 1B and 3) and impacted the distribution of several other histone modifications tested (Figs 3 and S7 and S8). Most prominently, we found that H3K36me3 was lost from most regions where it co-localized with H3K27me3 in wild type. The same regions lost H3K36me3 in the *ash1* deletion mutant, indicating that Ash1 activity is compromised in the *kmt5* mutant. Loss of Ash1, however, does not impact H4K20me3, indicating that H4K20me3 is epistatic to, or acts upstream of,

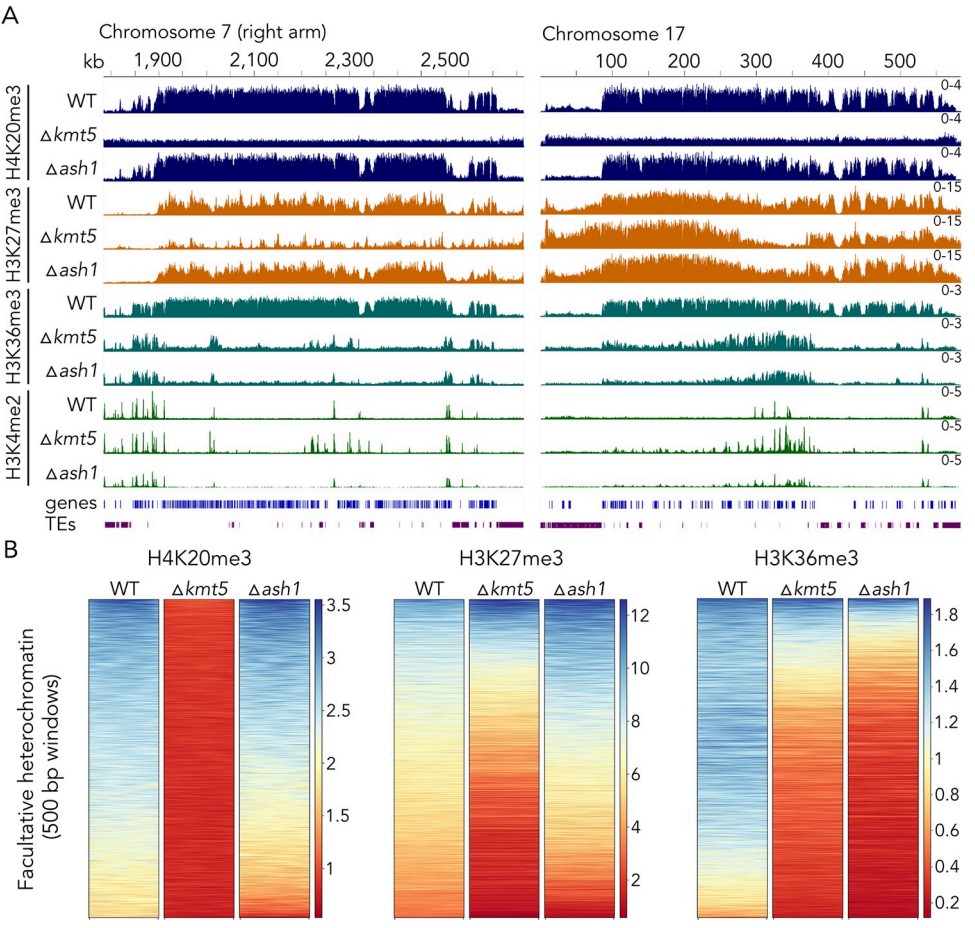

**Fig 3. Effects of deletion of *kmt5* or *ash1*, encoding two histone methyltransferases involved in the regulation of H4K20me3 and H3K36me3 levels. A)** In the absence of Kmt5, all H4K20 methylation is lost but H4K20me3 enrichment is not affected by lack of Ash1. Lack of Kmt5 or Ash1 alters H3K27me3 in selected regions of the genome, and the lack of either Kmt5 or Ash1 results in loss of H3K36me3 from regions that are not methylated by a second H3K36 methyltransferase, Set2. H3K36 methylation by Set2 is largely correlated with H3K4me2. These results strongly suggest that Kmt5 controls at least some H3K36 methylation, which in turn controls H3K27me3 in some regions of the genome. **B)** Comparison of enrichment of histone modifications in regions of facultative heterochromatin. Deletion of *kmt5* results in loss of H4K20me3, overall reduction in H3K27me3, and strong reduction in H3K36me3. Deletion of *ash1* affects H4K20me3 and H3K27me3 only mildly but strongly reduces H3K36me3 in these regions. Windows in the heatmap are ordered from highest enrichment (blue) to lowest enrichment (red) in wild type (WT).

Ash1-mediated H3K36me3. To characterize the specific role of Ash1 in mediating H3K36me, we also set out to generate deletion mutants of the main H3K36me methyltransferase, Set2. We obtained single Δ*set2* mutants that showed severe growth defects when compared to wild type (S9 Fig), but we were unable to obtain mutants for this methyltransferase in the Δ*ash1* background, suggesting that the presence of at least one H3K36 methyltransferase is required in *Z. tritici*. The severe growth defect of the Δ*set2* mutant did not allow us to perform ChIP-seq experiments that would have resulted in meaningful comparisons. We performed peak calling to quantify how much of the genome-wide H3K36me3 is mediated by Ash1 and Set2. Based on comparisons of wild type and Δ*ash1* H3K36me3 peaks, ~10% of H3K36me3 in wild type is not detectable in Δ*ash1*, indicating that Set2 is responsible for the vast majority (presumably ~90%) of H3K36me3, which aligns with previous reports in *N. crassa* [30]. However, this number may not fully represent Ash1-mediated H3K36me3, as in the absence of Ash1, Set2 may move to regions that are usually targeted by Ash1 in wild type. While Ash1-mediated H3K36me3 comprises less than 5% of H3K36me3 on core chromosomes (except for chromosome 7), Ash1 seems to be responsible for 50–95% of H3K36me3 on accessory chromosomes (S10 Fig and S1 Table). H4K20me3 overlaps with both, Set2-mediated H3K36me3 and Ash1-mediated H3K36me3, but we found a stronger correlation between Ash1-mediated H3K36me3 (~98% of domains overlap) than Set2-mediated H3K36me3 (~80% of domains overlap).

While impacts on H3K36me3 were similar in *kmt5* or *ash1* deletion strains, loss of *kmt5* had more severe effects on H3K27me3 enrichment. In the absence of Kmt5, we observed major changes in H3K27me3 distribution, including losses of large blocks of H3K27me3, movement to constitutive heterochromatin, and losses and gains of numerous shorter dispersed peaks. In addition, loss of H4K20me3, H3K36me3, and H3K27me3 in the Δ*kmt5* mutant was often accompanied by gain of H3K4me2, suggesting a transition from facultative heterochromatin to transcriptionally active euchromatin (Fig 3A). In contrast, we detected fewer differences in H3K27me3 enrichment in the *ash1* mutant, mostly losses or gains in a limited number of shorter dispersed peaks and increased enrichment in large H3K27me3-enriched blocks on accessory chromosomes.

We found that most changes in chromatin states in Δ*kmt5* and Δ*ash1* mutants occurred in regions that we characterized as facultative heterochromatin (cluster 1), i.e., regions that show enrichment for H3K27me3, H3K36me3, and H4K20me3, but to a much lesser extent in regions outside of facultative heterochromatin (S7 Fig). Looking specifically at enrichment of H3K27me3, H3K36me3, and H4K20me3 in regions of facultative heterochromatin (Fig 3B), we found that: H4K20me3 was absent in Δ*kmt5*, as expected, but only slightly reduced in Δ*ash1*; H3K27me3 was reduced in many regions in Δ*kmt5* but slightly increased in Δ*ash1*; and H3K36me3 was greatly reduced in Δ*ash1* and only slightly less reduced in Δ*kmt5* when compared to enrichment in wild type. Taken together, we showed that Kmt5 is important for facultative heterochromatin assembly and maintenance and that it is important for Ash1 activity.

## H4K20 mutations alter abundance and distribution of H4K20me3

To test whether the presence of the Kmt5 protein or its catalytic activity, i.e., methylation of H4K20, is important for H3K36me3, we constructed histone H4K20 mutants, replacing lysine residue 20 with either leucine (L, as a non-modifiable, relatively large residue), methionine (M, as a potential inhibitor of methyltransferase activity) or glutamine (Q, to mimic an acetylated lysine). The genome of *Z. tritici* encodes a single histone H4 (*hH4*) gene. We generated two different types of mutants: (1) ectopic addition of a mutant (or wild-type) histone allele, generating the potential for a mixed population of nucleosomes, and (2) ectopic addition of a mutant (or wild-type) histone allele in combination with deletion of endogenous *hH4* (Fig 4). The

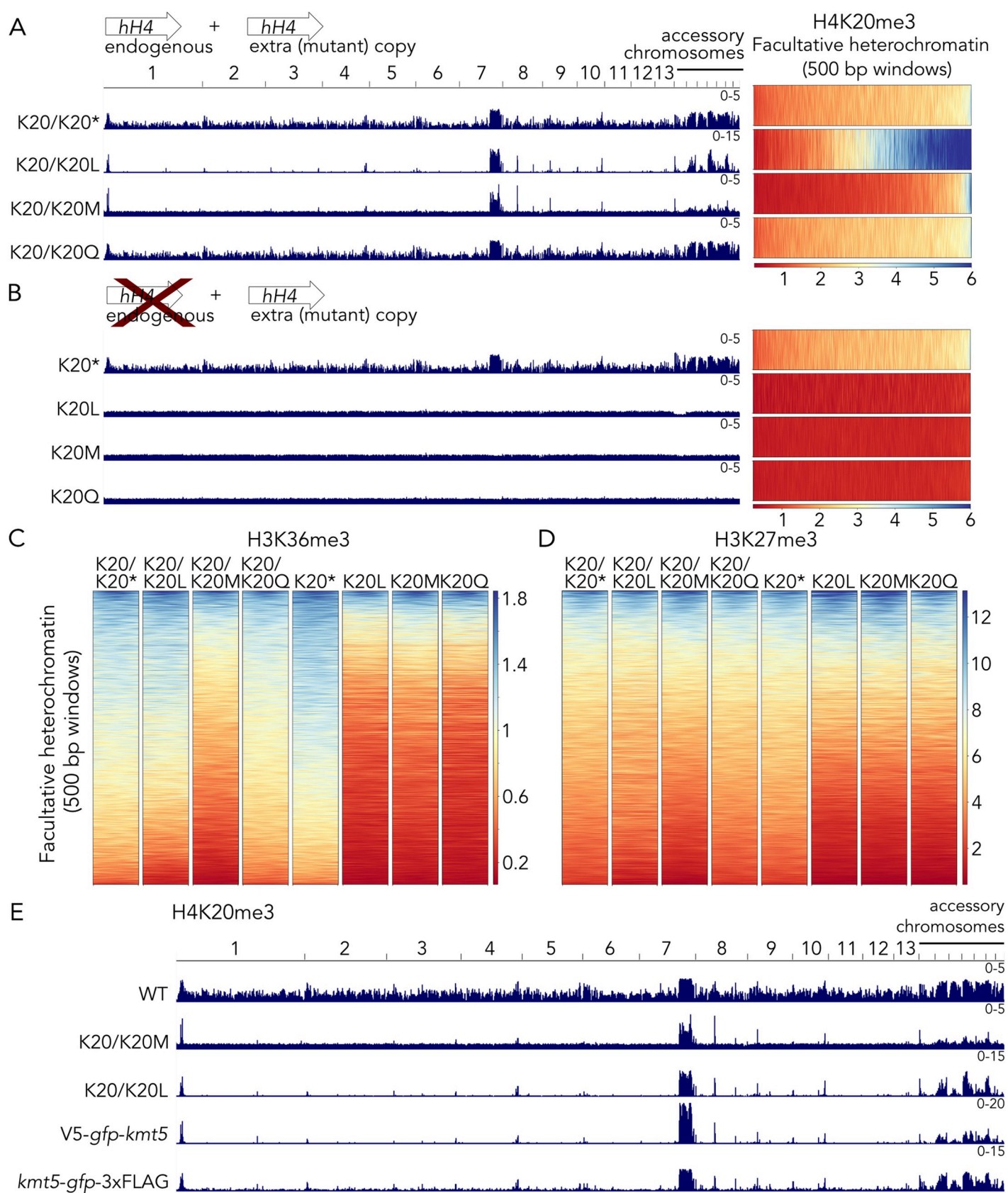

**Fig 4. H4K20 substitutions phenocopy deletion of *kmt5* but heterozygous H4 allele strains show specific H4K20me3 in regions of facultative heterochromatin in wild type. A)** H4K20me3 distribution in histone mutants with an additional *hH4* allele. Integration of another wild type allele (K20/K20) is used as a control strain. Integration of K20L (K20/K20L) and K20M (K20/K20M) alleles results in drastic changes in H4K20me3 distribution, while an additional K20Q allele (K20/K20Q) does not seem to affect H4K20me3 distribution. Data are normalized to RPGC (Reads Per Genomic Content; 1x normalization), the K20/K20L mutants showed higher relative enrichment than the other strains (see different scales). **B)** Deletion of the endogenous *hH4* gene in the wild type or mutant allele backgrounds results in absence of H4K20me3 in all mutant allele strains (K20L, K20M, K20Q) but normal distribution when the wild type allele is retained (K20). **C)** H3K36me3 is severely reduced in all strains with only a mutant allele (K20L, K20M, K20Q) and the K20/K20M strain, where H4K20me3 is reduced in facultative heterochromatin. **D)** H3K27me3 is mildly affected in the same strains that showed H3K36me3 reduction. **E)** A similar effect on H4K20me3 observed with heterozygous K20L (K20/K20L) and K20M (K20/K20M) alleles is found when Kmt5 is tagged with GFP on either the amino or carboxy terminus. *These strains represent control ("wild-type") strains in this experiment and contain an additional wild-type *hH4* (K20/K20) or deletion of endogenous *hH4* with an additional wild-type *hH4* gene (K20).

replacement alleles (no additional copies of *hH4*) recapitulated results obtained with the Δ*kmt5* mutant, namely complete absence of H4K20me3, loss of Ash1-mediated H3K36me3, and reduced H3K27me3 enrichment, revealing that the presence of the histone mark, but not necessarily the presence of the Kmt5 protein, is essential for Ash1 recruitment (Figs 4B–4D and S8). In contrast, we observed striking differences in H4K20me3 distribution in some of the strains with ectopic mutated *hH4* alleles. While the mutant with the K20Q (K20/K20Q) allele did not show differences from wild type, mutants with the K20L (K20/K20L) and K20M (K20/K20M) alleles completely lost H4K20me3 outside of facultative heterochromatin but retained H4K20me3 in some regions of facultative heterochromatin. The K20/K20L mutant showed strong enrichment in these regions, while the K20/K20M mutant showed overall reduced enrichment of H4K20me3 (Fig 4A). Reduction in H4K20me3 in the K20/K20M mutant also correlated with reduced H3K36me3 and H3K27me3 enrichment (Fig 4C and 4D). Presence of H4K20me3 in facultative heterochromatin in K20/K20L and K20/K20M mutants suggests that these regions are preferential targets of Kmt5. Based on data collected with mutant histone H4 alleles, presence of H4K20me3 is important for the deposition of H3K36me3 and, in some regions, H3K27me3 and thus for maintenance of a facultative heterochromatin state.

## Tagging of Kmt5 impairs protein function and limits H4K20me3 spreading

To characterize Kmt5 in more detail, we generated strains harboring N-terminally (*V5-gfp-kmt5*) or C-terminally (*kmt5-gfp-3xFLAG*) tagged Kmt5 alleles by either replacing the endogenous *kmt5* in wild type or complementing the Δ*kmt5* mutant at the original locus. Based on ChIP-seq analyses, none of these mutants retained or restored wild-type levels of H4K20me3 enrichment. H4K20me3 levels were similar to those observed in the K20/K20L and K20/K20M strains in specific locations co-localizing with Ash1-mediated H3K36me3 and H3K27me3 but mostly absent from the rest of the genome (Fig 4E). The C-terminally tagged Kmt5 mutant maintained some H4K20me3 outside of facultative heterochromatin but not at levels comparable to those in wild type. Notably, integration of the C-terminally tagged Kmt5 in either wild-type or Δ*kmt5* backgrounds led to similar defects in H4K20me3 distribution. This suggests the difference in H4K20me3 is not a result of impaired *de novo* deposition of H4K20me3 after complete loss of the modification but rather a defect in H4K20me3 genome-wide deposition or maintenance associated with impaired function of the C-terminally tagged Kmt5 protein (S8 Fig).

## Eaf3 is important for Ash1 activity

To further investigate the connection between H4K20me3 and Ash1-mediated H3K36me3, we searched the genome for predicted proteins that, in other organisms, have been shown to bind H4K20me3 or to interact with either Ash1 or Kmt5. In *S. pombe*, a PWWP domain-containing protein (Pdp1) occurs in a complex with Kmt5 and is required for all H4K20me [39]. We

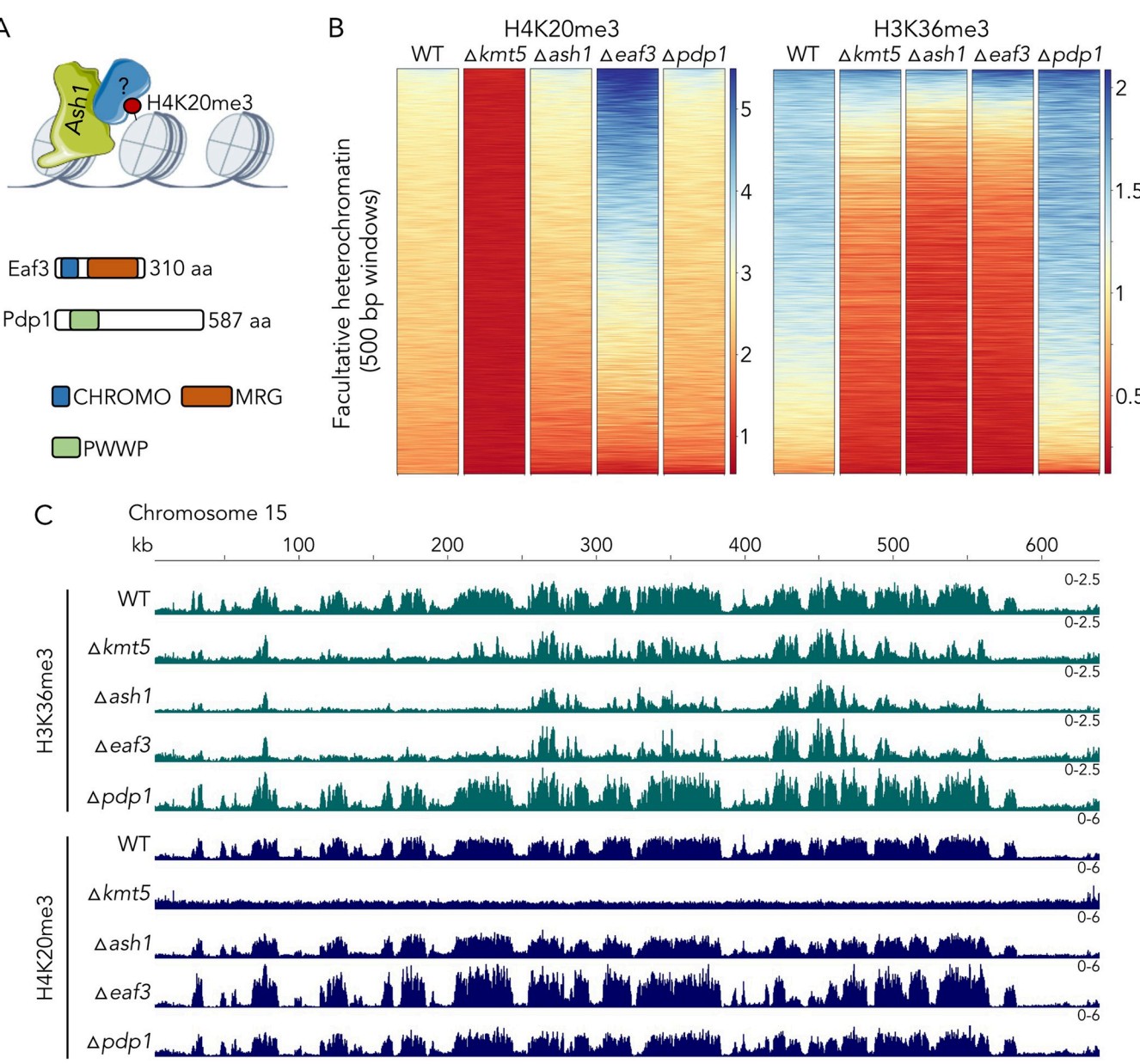

**Fig 5. Eaf3 links H4K20 and H3K36 methylation. A)** A putative H4K20me3-binding protein ("?") attracts the H3K36 methyltransferase, Ash1, to H4K20me3-enriched regions. The PWWP motif protein, Pdp1, is essential for H4K20me in *S. pombe*, while the CHROMO and MRG domain protein, Eaf3, had been shown to bind H3K36me3 in multiple organisms. **B)** Enrichment of H4K20me3 and H3K36me3 in facultative heterochromatin. We excluded the putative Pdp1 homolog (*pdp1*) because deletion of this gene had no effect on H4K20 or H3K36 methylation when compared to wild type (WT). The homologue of Eaf3, however, resulted in a large increase in H4K20me3, and a loss in H3K36me3, suggesting that it may interact with both H4K20me3 and H3K36me3. **C)** ChIP-seq tracks showing H3K36me3 and H4K20me3 on accessory chromosome 15 in WT and Δ*kmt5*, Δ*ash1*, Δ*eaf3*, and Δ*pdp1* strains.

found a single protein with a PWWP domain in *Z. tritici* (Zt_chr_4_873) and other filamentous fungi (Fig 5A). Deletion of the respective gene in *Z. tritici* did not affect H4K20me3 or H3K36me3 (Fig 5B and 5C) but rather led to increased enrichment of H3K27me3 in numerous regions along the genome (S7, S8 and S11 Figs), suggesting that there is no PWWP domain-containing ortholog of the *S. pombe* Pdp1 in *Zymoseptoria*. A homolog of the *Z. tritici* Pdp1 (PWWP domain-containing protein 1) was found to interact with a chromatin

remodeling factor, ISWI, in *N. crassa*, where it also increased H3K27me3 levels [40]. Based on the similarity among the *Zymoseptoria*, *Neurospora*, and *Fusarium* Pdp1-like proteins, we propose that these proteins are orthologs of the *Saccharomyces cerevisiae* protein Ioc4 [40].

Our second candidate protein for connecting H4K20me3 and Ash1-mediated H3K36me3 was a homolog of the chromodomain-containing protein Eaf3 in *S. cerevisiae* or MRG15 in *D. melanogaster*. In *S. cerevisiae*, Eaf3 is part of the histone deacetylase complex Rpd3S [41] and the histone acetyltransferase complex NuA4 [42] and its chromodomain can bind both H3K36me3 and H4K20me3 *in vitro* [43], though not at the same time. In *D. melanogaster*, MRG15 interacts with Ash1 and is essential for Ash1 function [44]. We deleted the *Z. tritici* homolog of the *eaf3* gene (Zt_chr_1_1455) and performed ChIP-seq on the mutant. Absence of Eaf3 resulted in loss of H3K36me3 in the same regions where we observed loss of H3K36me3 in Δ*ash1* and Δ*kmt5* mutants indicating that Eaf3 is required for Ash1 activity (Figs 5B and 5C and S7 and S8). Eaf3 was not required for H4K20me3; on the contrary, we observed increased levels of H4K20me3 in the Δ*eaf3* mutant in regions that are targeted by Ash1 in wild type (Fig 5B and 5C). We did not observe a similar increase in H4K20me3 outside of regions of facultative heterochromatin. Because we did not detect a similar increase in H4K20me3 in Δ*ash1* mutants, it is likely not the absence of H3K36me3 but a mechanism upstream of Ash1 recruitment that influences H4K20me3 levels in Δ*eaf3*.

## H4K20me is required for transcriptional repression in facultative heterochromatin

To investigate whether the observed changes in chromatin structure correlate to changes in gene expression, we sequenced mRNA by enriching for poly(A)-containing transcripts from three replicates of wild-type, Δ*kmt5*, Δ*ash1*, Δ*eaf3*, hH4$^{K20M}$, and hH4$^{K20/K20M}$ strains and determined which genes were differentially expressed (Figs 6A and S12). As a control for the K20/K20M strain, we also included a strain that carries an ectopic wild-type *hH4* copy (K20/K20). We detected few (only 10) differentially expressed genes between wild type and this strain, concluding that an additional wild-type *hH4* copy has negligible impact under the conditions tested (S2 Table). Except for the Δ*kmt5* strain (no *kmt5* expression), we did not detect any significant differences in expression levels for *kmt5* in any of the mutant strains (S2 Table). We showed previously [33] that loss of H3K27me3 alone has surprisingly minor effects on transcription, as only a few genes, mostly located on accessory chromosomes, showed increased levels of expression. In contrast, deletion of *kmt5* resulted in upregulation of a large set of genes, including many genes in regions of facultative heterochromatin normally enriched with H3K27me3, Ash1-mediated H3K36me3, and H4K20me3. Out of 718 upregulated genes (DESeq2, padj ≤0.01, | log2 fold change | ≥0.585, Fig 6A) in Δ*kmt5*, 286 were located in facultative heterochromatin (~40% of upregulated genes). Similarly, 39% of upregulated genes in the K20M mutant were within facultative heterochromatin, indicating that Kmt5 and H4K20me3 are especially important for gene repression in those regions. While all mutant strains we included showed significant enrichment of upregulated genes in facultative heterochromatin (chi-square ≥0.001 for all mutants), the relative proportion differs (26% in Δ*ash1*, 19% in K20/K20M, and 15% in Δ*eaf3*). In these mutants, H4K20me3 levels were reduced in some regions, largely unchanged, or even increased (Δ*eaf3*) suggesting that the presence of H4K20me3 may limit derepression. We found that more than 55% of upregulated genes were shared in Δ*kmt5* and Δ*ash1* mutants and almost 90% of upregulated genes in the K20M mutant were also upregulated in Δ*kmt5*, Δ*ash1*, or both. In general, a large proportion of upregulated genes are shared in the mutants we tested, indicating that they affect a common silencing pathway (S12 Fig). Most of the upregulated genes (~94%), similar to genes in

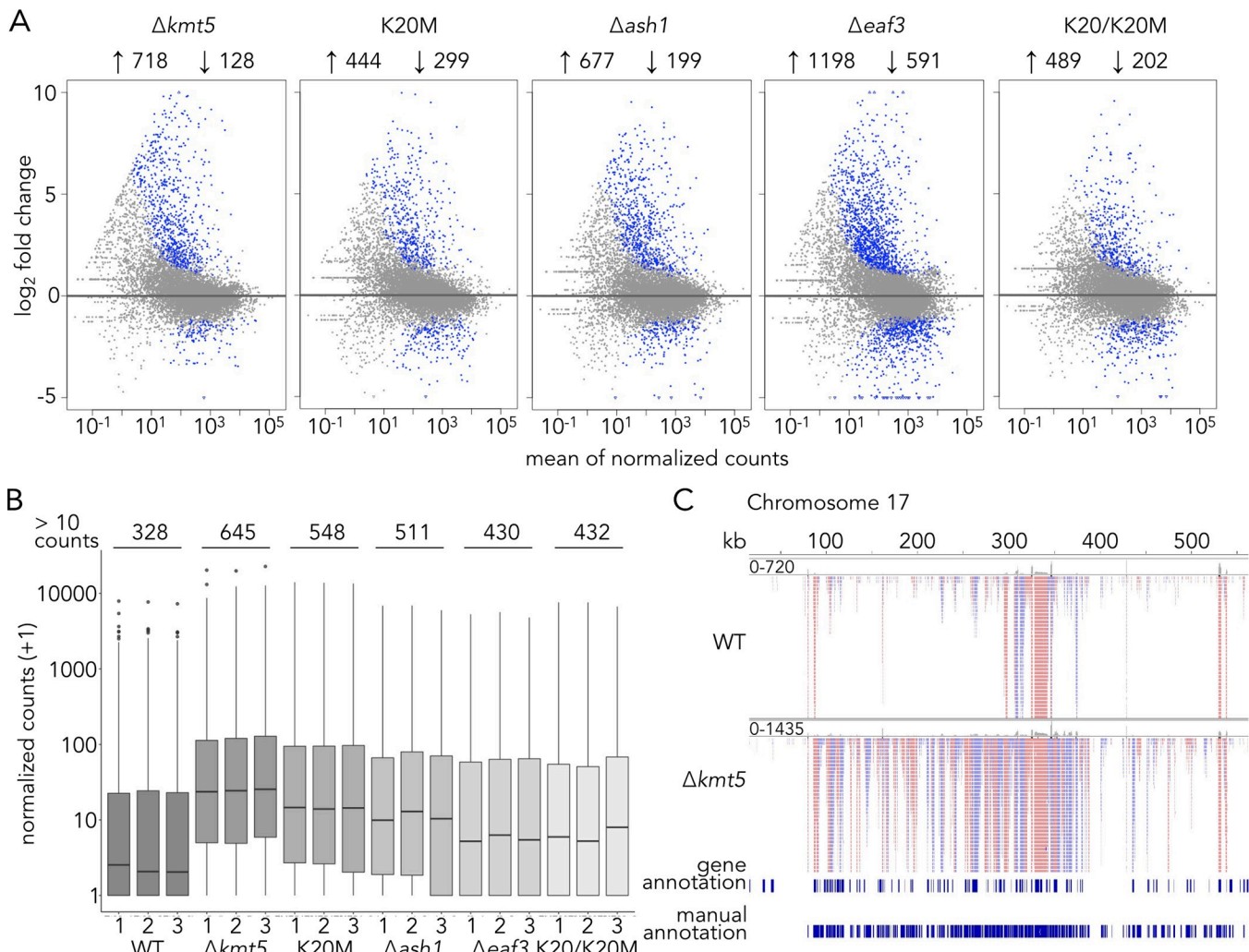

**Fig 6. H4K20 methylation is important for transcriptional silencing in facultative heterochromatin. A)** Results from differential expression analysis (DESeq2, padj ≤0.01, | log2 fold change | ≥0.585) comparing wild-type and mutant strains. All differentially expressed genes (within and outside of facultative heterochromatin) are shown in blue. **B)** Normalized counts assigned to transcripts in facultative heterochromatin (number of newly annotated transcripts per strain on top); transcripts were newly annotated with stringtie. RNA-seq results on *kmt5*, *eaf3*, and histone mutants, as well as on wild-type (WT) samples revealed a large group of novel transcripts, especially on accessory chromosomes, that had not been detected before and that come from H3K27me3-enriched regions. These transcripts were more highly expressed in most mutants compared to wild type, especially if H4K20me3 is absent. **C)** RNA-seq reads of wild type and Δ*kmt5* mapped to accessory chromosome 17. The previous annotation [77] identified only a fraction of transcripts expressed on accessory chromosomes.

facultative heterochromatin in general, lack functional annotations (only 57 out of 976 [~6%] genes have been previously annotated in these regions), making it difficult to assess enrichment for putative functions among these genes. A list including all genes, expression levels, and predicted functions can be found in the S2 Table.

We noticed that, especially on accessory chromosomes, many regions without previously annotated genes or with incorrectly annotated genes showed transcriptional activity, often in a strand-specific manner generating spliced, likely poly(A)-containing transcripts, which suggested transcription to generate mRNA by RNAPII. Many of these transcripts were highly or newly expressed in the Δ*kmt5* mutant (Fig 6C). To quantify the number of transcripts that were not captured by the current gene annotation, we used stringtie [45] to identify transcripts

**Table 2. Manual annotation of transcripts expressed on chromosome 17 (Fig 6C) based on combined RNA-seq data from wild type and mutants.**

|  | number of transcripts |
| --- | --- |
| identified | 186 |
| protein coding | 13 |
| hits on accessory chromosomes | 89 |
| hits on core chromosomes | 62 |
| hits in other (fungal) species | 41 |
| functional annotation | 31 |

in the combined RNA-seq data of all strains. We identified a total of 1,023 transcripts in facultative heterochromatin, of which only 516 were partially or fully overlapping with the previous annotation. Expression levels of these transcripts were highest in the Δ*kmt5* and K20M mutants followed by the Δ*ash1* mutant (Fig 6B). The Δ*eaf3* and K20/K20M mutants exhibited only slightly higher expression of these transcripts than wild type (Fig 6B). Many of the newly discovered transcripts do not appear to encode proteins, as they contain numerous, presumably premature, stop codons in all three reading frames. To begin characterization, we manually annotated all transcripts expressed on one accessory chromosome (chromosome 17). Their chromatin state (in wild type), presence of sequence similarity to other accessory or core chromosomes, any potential function based on blastx searches, and presence of homologs in other species are listed in Table 2 and S3 Table. We identified a total of 186 transcripts of which 55 match or partially match the previous annotation. Only 13 of these 186 transcripts seem likely to encode a protein based on the presence of an open reading frame without premature stop codons; one of these transcripts encodes a putative protease, while the others showed no predicted functional domains. If these 186 transcripts do not encode proteins, can we identify any characteristics that shed light on putative function or origin? To address this, we looked for sequence similarity on other accessory chromosomes or core chromosomes using blastn searches in the *Z. tritici* wild-type genome. Almost half of the transcripts (89/186) on chromosome 17 showed sequence similarity to other accessory chromosomes, often multiple hits on multiple chromosomes. Almost one third (62/186) showed similarity to sequences or genes on core chromosomes. This sequence similarity was mostly restricted to short regions of the transcript and the gene on the core chromosome and did not fully cover the complete open reading frame (S13 Fig). Among the hits on core chromosomes and functional annotation based on blastx searches, we found several genes that encode proteins involved in chromosome organization and segregation and cell cycle control (S3 Table). For example, we found eleven transcripts with similarity to the gene encoding a Shugoshin homolog (Zt_chr12_379), a protein that is important for chromosome cohesion during mitosis and meiosis (S13 Fig). These eleven transcripts show partial sequence similarity to different regions of the *shogushin* gene on the core chromosome. We do not know if these transcripts may act as regulatory RNAs, and genes with sequence similarity on core chromosomes were not differentially expressed in any of the mutants, except for four genes (three in Δ*eaf3* and one in the K20M mutant), which did not seem to correlate with the corresponding transcript levels from chromosome 17.

## Discussion

We report chromatin dynamics and interactions in facultative heterochromatin that identify Kmt5 and H4K20me3 as important regulators for transcriptional repression and recruitment of Ash1-mediated H3K36me3 in *Z. tritici*, which itself at least partially regulates the

distribution of H3K27me3. Here we investigated potential epistasis rules for repressive histone modifications and show that presence of H4K20me3 and Kmt5 are important for Ash1-mediated H3K36me3, because we observed absence of H3K36me3 in selected regions upon either deletion of Kmt5 or introduction of mutant histone alleles. Our findings suggest that H4K20 methylation needs to be established before Ash1 can catalyze H3K36me3.

Based on our ChIP-seq data, H4K20me3 is a widespread mark, heavily enriched along *Z. tritici* chromosomes, especially in regions enriched with H3K27me3, segments of the genome we refer to as facultative heterochromatin. Here, we elucidated the function of H4K20me3 in the formation and maintenance of facultative heterochromatin. Why H4K20me3 is also enriched across some gene bodies outside of facultative heterochromatin remains unclear. In fungi, Kmt5 catalyzes all three methylation states, which makes it difficult to disentangle the relative importance of the three different potential H4K20 methylation states.

In animals, H4K20 methylation is found in various chromatin environments in a tissue- and life stage-dependent manner [11]. Because of the presence of multiple H4K20 methyl-transferases, non-histone targets, and interdependency of methylation states, the distinct roles of the different methylation states are difficult to study. H4K20me1 has been associated with both active [24,46] and repressed genes [47]. It has been linked to promoting chromatin openness [24], but has also been shown to be enriched in inactive regions, such as the inactivated X-chromosome [48]. Previous studies found that H4K20me3 levels increase during senescence and in aged tissues but decrease in cancer cells, which is associated with increased and decreased silencing, respectively, and suggests that H4K20me3 is important in suppressing tumorigenicity [49,50]. The C-terminus of KMT5C has been shown to interact with HP1, targeting H4K20me3 to H3K9me3-enriched regions, such as TEs, telomeres, and pericentric regions [51,52]. This overlap of H4K20me3 and H3K9me3 appears to be absent in *Z. tritici* as H4K20me3 was not involved in silencing constitutive heterochromatin, suggesting that Kmt5 in this species has different interaction partners than in animals. In further contrast to animals, fungi only possess a single methyltransferase mediating all H4K20me, as opposed to different enzymes catalyzing either H4K20me1 or H4K20me2/3. The presence of different H4K20 HMTs may result in more specialized interactions with different complexes that are specific to the different states of H4K20 methylation.

We found that H4K20me3 overlaps with Ash1-mediated H3K36me3 and H3K27me3 in facultative heterochromatin but also co-occurs with H3K36me3 outside of these regions. An overlap of H4K20me3 and H3K36me3 in gene bodies of actively transcribed genes has been shown in embryonic stem cells, possibly resembling the mechanisms that are responsible for H4K20me3 enrichment outside of facultative heterochromatin regions found here [53]. Very little is currently known about the connection between H4K20me and H3K27me3. H4K20me3 and H3K27me3 have been shown to co-occur on some transposons in *Xenopus tropicalis* embryos [54] and similar mechanisms for recruitment to X-chromosomes have been proposed for H4K20me1 and H3K27me3 [48]. Results from other fungi are sparse, as no genome-wide studies have been carried out, even in model organisms such as *S. pombe* and *N. crassa*. Lastly, H4K20 methylation appears to be lacking completely in plants [55], suggesting that facultative heterochromatin is regulated in different ways in different organisms.

To understand the biochemistry of a putative Kmt5 complex, we tagged the protein with GFP or short tags for protein purification. We showed that both N- and C-terminal tags greatly interfered with normal Kmt5 function, as documented by ChIP-seq. Taken together with our results on the added histone H4 alleles (K20/K20L and K20/K20M), facultative heterochromatin may be the first region in which H4K20 methylation occurs, and from which this histone mark spreads into the bulk of the chromatin to normal levels observed in wild type. It is unclear how this partial activity occurs, but studies on other histone mutants, specifically

H3[K27M] showed that K to M mutations alter the release of histone methyltransferases from their substrates and thus sequester them on chromatin [56,57]. The consequence is limited or greatly reduced activity of the methyltransferase in other parts of the genome, which is what we observed with the K20/K20L and K20/K20M mutants. The reason N- and C-terminal Kmt5 fusion proteins show similar distribution may well relate to the overall structure of the Kmt5 protein, with the N-terminus being slightly more sensitive to modifications or additions than the C-terminus. In the absence of fully, or at least largely, complementing tagged alleles of Kmt5, we will address Kmt5 biochemistry in a separate study.

How then does H4K20 trimethylation govern facultative heterochromatin *de novo* assembly and maintenance? Our current working model based on the genetic data presented here (Fig 7) proposes that in maintenance mode Kmt5, alone or in a complex, methylates H4K20 in regions that are already enriched with methylated H4K20. For *de novo* methylation, some "signal" must be present. When Kmt5 is reintroduced into a Δ*kmt5* mutant, the protein, alone or in a complex, is able to detect such signals and methylate H4K20, as observed in the tagged and complemented Kmt5 strains we generated. These signals do not appear to be other histone modifications we analyzed in this study (H3K27me3, Ash1-mediated H3K36me3), as none of the mutants, except for Δ*kmt5* and histone H4 mutants, showed reduced H4K20me3 levels in facultative heterochromatin.

Independent of how Kmt5 first finds its targets, it seems possible that H4K20me3 is bound by Eaf3 which then stimulates activity of Ash1, as shown for MRG15 in flies [44,58]. This is supported by finding increased H4K20me3 levels in the absence of Eaf3, which may be a result of continued methylation of H4K20 by Kmt5 in the absence of Eaf3 binding to H4K20me3 and the requirement of Eaf3 for Ash1-mediated H3K36me3. Whether Eaf3 is part of an Rpd3S-like complex in filamentous fungi or whether it may directly recruit Ash1 to regions of H4K20me3 remains unknown. We propose that Ash-1-mediated H3K36 methylation results in a switch of Eaf3 binding H4K20me3 to H3K36me3, though it is also possible that Eaf3 may bind both modifications at the same time.

Little is known about proteins that interact with Kmt5 in fungi except for *S. pombe* Pdp1 [39]. Here we showed that there is no true ortholog for this protein in *Z. tritici* and that the best homolog may be an ortholog of PWWP domain-containing protein Ioc4 of *S. cerevisiae*, a component of some ISWI chromatin remodeling complexes in *S. cerevisiae* and *N. crassa* [40,59,60]. While *Z. tritici* Pdp1 does not seem to be important for H4K20me3, we did observe increased enrichment of H3K27me3 outside of wild-type facultative heterochromatic regions in the absence of *Z. tritici* Pdp1, suggesting that it has an important role in chromatin regulation; modest increases of H3K27me3 enrichment were also observed in the absence of the orthologous protein in *N. crassa* [40].

Similar to findings in *N. crassa* and *F. fujikuroi* but unlike in animals, H3K27me3 and Ash1-mediated H3K36me3 occur in the same regions [28,30,31,37]. Loss of Ash1-mediated H3K36me3 has minor impacts on H3K27me3 in *Z. tritici*, resulting in increased enrichment in some regions and decrease in other regions, which is similar to findings in *N. crassa* where H3K27me2/3 was impacted by a catalytically inactive Ash1 protein [30]. While Ash1 in animals is predominantly an H3K36me2 methyltransferase, we found major impacts on H3K36me3, as shown previously in *N. crassa* and *F. fujikuroi* [30,31]. Fungal Ash1 proteins are different from Ash1 in animals, in that they lack C-terminal Bromo, PHD, and BAH domains, and, thus, may have different chromatin binding properties and/or functions than their animal homologs.

Functional analyses of the Kmt5 homolog in *S. pombe*, Set9, and some of the animal Kmt5 proteins suggest importance during replication and DNA repair [15,16,20,21,61,62]. Here, we showed that both Kmt5 and Ash1 are necessary for normal growth of *Z. tritici* but not essential

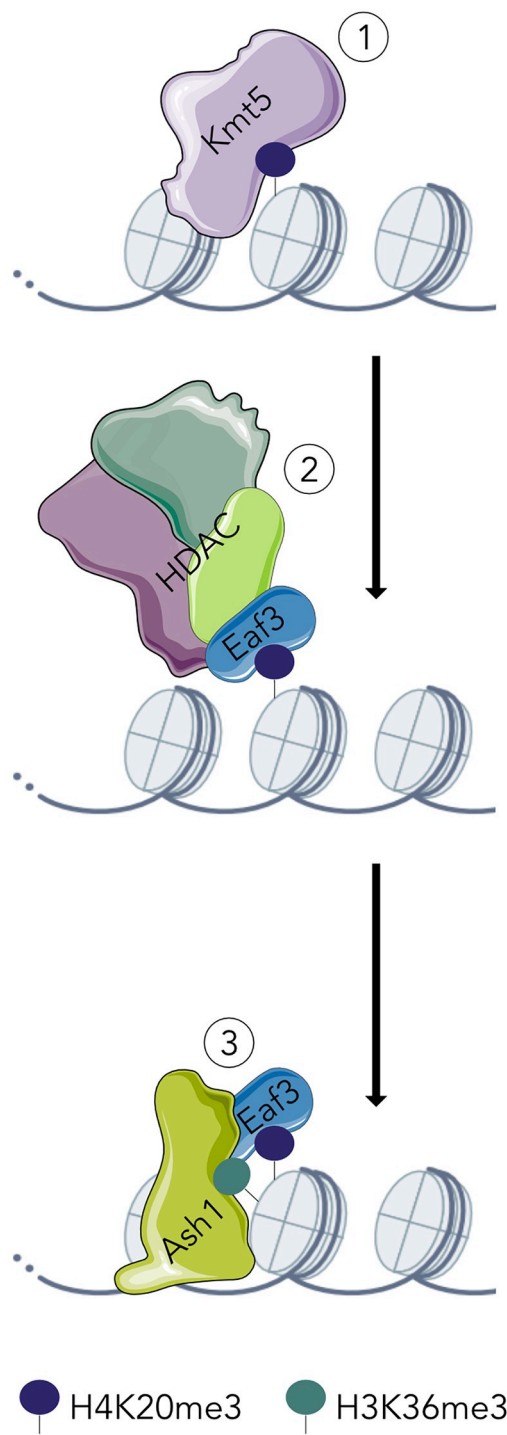

**Fig 7. Working model for how H4K20me3 may direct Ash1-mediated deposition of H3K36me3.** (1) Kmt5 mediates H4K20me3 followed by (2) binding of Eaf3 to H4K20me3; Eaf3 may be in a complex with a histone deacetylase (e.g., RPD3S). (3) Presence of Eaf3 promotes Ash1-mediated H3K36me3.

for viability. While the Δ*ash1* mutant showed only minor effects under our conditions, the Δ*kmt5* mutant was more sensitive to genotoxic stress, especially at higher temperatures. Lack of either Kmt5 or Ash1 also resulted in accelerated chromosome loss rates; while the loss rate for wild type was ~3%, mirroring our earlier results [33,38], the loss rate was 41% and 58% in Δ*ash1* and Δ*kmt5*, respectively. Increased chromosome loss affected some chromosomes more than others. Chromosomes 14 and 16 are easily lost even in wild-type strains, especially under heat stress conditions [33,36,38], and they were also lost most commonly, by far, in Δ*kmt5* and Δ*ash1*. Thus, unlike the loss of the H3K27me3 methyltransferase in *Z. tritici*, Kmt6, which results in an overall more stable genome [33], lack of Ash1 or Kmt5 resulted in decreased genome stability, which suggests different, separable functions in genome maintenance and control of gene expression in facultative heterochromatin for both Ash1 and Kmt5.

Our earlier studies showed that lack of H3K27me3 alone did not result in large-scale upregulation of regions enriched with H3K27me3 [33]. One of our goals was to uncover which *Z. tritici* histone modifications, if any, take on the gene silencing function usually mediated by H3K27me3 in other organisms. Here, we showed that H4K20me3 and Ash1-mediated H3K36me3 are important for transcriptional repression of facultative heterochromatin. Some of these effects are phenocopied by replacement of histone H4K20 with a methionine (K20M), or in a strain with a wild type and H4K20M allele (K20/K20M). While some *bona fide* genes on core chromosomes and even accessory chromosomes are affected by the lack of Kmt5 or Ash1, most regions enriched with H3K27me3, Ash1-mediated H3K36me3, and H4K20me3 have relatively few previously annotated genes. To understand the transcripts that emanate from these regions we manually curated RNA-seq data for one accessory chromosome. We found that most transcripts are expressed from regions that do not appear to encode canonical genes. The presence of multiple copies on different accessory chromosomes and the fragmented sequence similarity, as shown for the *shogoshin* gene, may hint to ancestral duplication events of core genes on accessory chromosomes followed by mutation and frequent structural rearrangements. How exactly these RNA species act, if they have specific functions, how they arose, why they are expressed, and whether they interfere with or aid the propagation of accessory chromosomes in *Z. tritici* are interesting questions but beyond the scope of the current study. Our results show, however, that studies resulting in a lack of specific histone modifications may necessitate renewed annotation efforts, which should capture transcripts from pseudogenes and apparent *bona fide* genes with few nonsense codons.

In conclusion, we showed that H4K20me3, H3K27me3, and Ash1-mediated H3K36me3 together delineate facultative heterochromatin in *Z. tritici*. H4K20me3, as well as the presence of Eaf3, are required for normal Ash1-mediated H3K36me3, uncovering a novel mechanism for the establishment and maintenance of facultative heterochromatin and transcriptional silencing.

## Methods

### Fungal and bacterial growth conditions

*Zymoseptoria tritici* cultures were grown on YMS (4 g yeast extract, 4 g malt, 4 g sucrose per liter, with 16 g agar per liter added for solid medium). Glycerol stocks (1:1 YMS and 50% glycerol) were maintained at -80˚C. *Escherichia coli* strain DH10beta (NEB) was grown at 37˚C in LB (10 g tryptone, 10 g NaCl, 5 g yeast extract per liter) with appropriate antibiotics (100 μg/mL kanamycin). *Agrobacterium tumefaciens* strain AGL1 was grown in dYT (16 g tryptone, 5 g NaCl, 10 g yeast extract per liter) or LB media with appropriate antibiotics (100 μg/mL kanamycin, 100 μg/mL carbenicillin, 50 μg/mL rifampicin).

## Generation of plasmids for fungal transformation

Plasmids were generated by Gibson Assembly [63] using a homemade Gibson reaction mix based on recipes from OpenWetWare [64] (5x Isothermal Reaction Mix [600 μL]: 300 μL 1 M Tris-HCl [pH 7.5], 30 μL 1 M MgCl$_2$, 6 μL 100 mM dGTP, 6 μL 100 mM dATP, 6 μL 100 mM dTTP, 6 μL 100 mM dCTP, 30 μL 1 M DTT, 0.15 g PEG 8000, 60 μL 50 mM NAD$^+$, 156 μL H$_2$O) and modified by using the "enhanced" version [65] (1.33x Gibson-Assembly Master-Mix [for 25 reactions]: 100 μL 5x Isothermal Reaction-Mix, 215.32 μL H$_2$O, 0.2 μL T5 Exonuclease [10 U/μL NEB #M0363], 6.25 μL Phusion DNA-Polymerase [2 U/μL ThermoFisher #F530L], 3.125 μL ET SSB [500 μg/ml NEB #M2401S], 50 μL Taq DNA-Ligase [40 U/μL stock NEB #M0208L]). Primers, plasmids, and strains used in this study are listed in S4 Table.

## Transformation of *Zymoseptoria tritici*

Transformation was carried out by *A. tumefaciens* [66]. In our updated protocol (detailed protocol is provided as S1 Method), *A. tumefaciens* strain AGL1 (transformed with the respective target plasmid, see S4 Table) was grown for ~24 h in liquid culture (LB with 100 μg/mL kanamycin, 100 μg/mL carbenicillin, and 50 μg/mL rifampicin) at 28˚C before induction. Induction was performed by centrifuging LB cultures (3000rpm for 5 min), and resuspending cells in 1 mL of induction medium (10 mM K$_2$HPO$_4$, 10 mM KH$_2$PO$_4$, 2.5 mM NaCl, 2 mM MgSO$_4$, 9 μM FeSO$_4$, 40 mM MES buffer, 0.7 mM CaCl$_2$, 0.5% glycerol, 5 mM glucose, 6.25 mM NH$_4$NO$_3$, 0.002% Vogel trace elements, 100 μg/mL kanamycin, 100 μg/mL carbenicillin, 50 μg/mL rifampicin, 200 μM acetosyringone) and inoculating resuspended AGL1 cultures in 5 mL induction medium (OD$_{590}$ = 0.15) for overnight growth at 28˚C. Induced cultures were adjusted to OD$_{590}$ = 0.4 the following day and mixed with *Z. tritici* cells (OD$_{590}$ = 1) at a 1:1 ratio. To grow recipient *Z. tritici* cells for this step, spores were streaked out on YMS plates from glycerol stocks 3–4 days before transformation, resuspended in water, and the concentration adjusted to an OD$_{590}$ = 1 in induction medium, before mixing with induced AGL1 cells. Aliquots of 250 μL of the *A. tumefaciens* AGL1 and *Z. tritici* mixture was plated on Hybond N + membranes on plates with induction medium and incubated at room temperature for 48–72 h. Membranes were transferred to YMS plates containing the appropriate antibiotics for selection (250 μg/mL cefotaxime, 150 μg/mL timentin, 100 μg/mL G418, 100 μg/mL hygromycin) and incubated for another ~10 days at 18˚C until resistant colonies appeared.

## Chromatin immunoprecipitation and sequencing (ChIP-seq)

ChIP experiments were carried out as described previously [67] with some modifications; a detailed protocol is provided as S2 Method. Briefly, fungal spores were grown on YMS plates for three to four days, harvested, and resuspended in 5 mL of 1x PBS (137 mM NaCl, 2.7 mM KCl, 10 mM Na$_2$HPO$_4$, 1.8 mM KH$_2$PO$_4$). Cells were crosslinked using 0.5% formaldehyde for 15 min at room temperature and quenched by adding 150 μL of 2.5 M glycine. Crosslinked cells were centrifuged at 3,000 rpm for 5 min at 4˚C and washed once with cold 1x PBS. We found that grinding cell pellets to a fine powder in liquid nitrogen and storage at -80˚C resulted in no noticeable decrease in ChIP efficiency (storage for at least six months was acceptable). For ChIP, frozen, ground cells were resuspended in ice-cold ChIP lysis buffer (50 mM HEPES-NaOH [pH 7.5], 90 mM NaCl, 1 mM Na-EDTA [pH 8.0], 1% Triton X-100, 0.1% DOC, and proteinase inhibitors [1 mM PMSF, 1 μg/mL Leupeptin, 1 μg/mL E-64 and 0.1 μg/mL Pepstatin]) in a ratio of 5 μL ChIP lysis buffer to 1 mg of ground cells. To 1 mL of lysate, 2 μL 1 M CaCl$_2$ and 5 μL micrococcal nuclease (NEB, #M0247S) were added and chromatin was fragmented into predominantly mono-nucleosomes by incubation at 37˚C for 15 min. The reaction was stopped by adding 20 μL of 0.5 M Na-EGTA, pH 8. Digested chromatin was centrifuged at

6,000 rpm for 5 min and 250–300 μL of the supernatant was used per ChIP reaction. Chromatin was pre-cleared by incubating with protein A Dynabeads (Invitrogen) at 4°C for 1 h followed by overnight incubation with 3 μL of the following antibodies: H3K4me2 (Millipore 07–030), H3K9me3 (Active Motif 39161), H3K27me3 (Active Motif 39155), H3K27me2 (Active Motif 39245),) H3K36me3 (Active Motif 61101; abcam 9050), H4K20me3 (Active Motif 91107). After overnight incubation, 25 μL protein A Dynabeads were added and samples were incubated for 1–2 h at 4°C. Beads were washed twice with ChIP lysis buffer, once with ChIP lysis buffer + 0.5 M NaCl (50 mM HEPES-NaOH [pH 7.5], 500 mM NaCl, 1 mM Na-EDTA [pH 8.0], 1% Triton X-100, 0.1% DOC), once with LiCl wash buffer (10 mM Tris-HCl [pH 8.0], 250 mM LiCl, 0.5% IGEPAL CA-630, 0.5% DOC, 1 mM Na-EDTA [pH 8]) and once with 1x TE buffer. Chromatin was eluted in 125 μL TES (50 mM Tris-HCl [pH 8.0], 10 mM Na-EDTA [pH 8], 1% SDS) and de-crosslinked overnight at 65°C. Samples were treated with RNAse A (2 h at 50°C) and protein-ase K (2 h at 65°C) and DNA was extracted with the ChIP DNA Clean & Concentrator kit (Zymo Research). Libraries for sequencing were prepared with a modified version of the Next Ultra II DNA Library Prep Kit for Illumina (NEB, #E7645S), see S3 Method. Sequencing was performed on an Illumina HiSeq 3000, obtaining 2 x 150-nt read pairs by Admera Health (South Plainfield, NJ, USA).

## RNA isolation and sequencing (RNA-seq)

Fungal cells were grown and harvested as described for the ChIP experiments. Cells were ground in liquid nitrogen and ~100 mg of ground cells were used for total RNA extraction by a TRIZOL (Invitrogen) method. Extraction was carried out according to the manufacturer's instructions and total RNA was treated with DNAse I. Stranded mRNA libraries were prepared by a modified version of a previously published protocol [68]; the detailed protocol is provided as S4 Method. Stranded mRNA libraries were sent to Admera Health (South Plainfield, NJ, USA) for sequencing on an Illumina HiSeq 3000, obtaining 2 x 150-nt read pairs.

## Quantification of chromosome loss

Each strain (wild type, Δkmt5, Δash1) was grown in five replicate populations on YMS plates, and every three to four days a fraction of the population was transferred to a new plate. After eight transfers (four weeks), populations were diluted and spread on YMS plates to obtain single colonies, representing individual cells of the respective population. A total of 48 colonies/replicate (i.e., 240 per strain) were screened by PCR for the presence of all accessory chromosomes as previously described [38].

## Western blots

Fungal cells were grown on YMS plates for three to four days, harvested and washed in 1x PBS. Cells were resuspended in three volumes of high-salt extraction buffer (25 mM HEPES-NaOH [pH 7.9], 700 mM NaCl, 0.1 mM Na-EDTA [pH 8.0], 0.2% NP-40, 20% glycerol, 1.5 mM MgCl₂, 1 mM PMSF, 1 mM DTT, 1 μg/mL Leupeptin, 0.1 μg/mL Pepstatin) and incubated on ice for 10 min. Samples were sonicated (Branson Sonifier-450) for three sets of 10 pulses (output = 2, duty cycle = 80), keeping the sample on ice between sets. Insoluble material was removed by centrifugation at 14,000 rpm and 4°C for 10 min. Protein concentrations were quantified with a Qubit fluorimeter (Invitrogen). Total protein (75 μg per lane) was loaded onto pre-cast 4–20% gradient Mini-PROTEAN TGX Stain-Free SDS PAGE gels (BioRad). Proteins were transferred onto nitrocellulose membrane (0.2 μm pore size; Bio-Trace) by wet transfer (1 h, 110V). The following primary antibodies were used: αH3 (Active motif 39763, mAb, 1:5000), αH4K20me1 (abcam 9051, pAb, 1:1000), αH4K20me3 (Active

motif 39180, pAb, 1:2000). The following secondary antibodies were used: IRDye 680RD Goat anti-Mouse IgG (LI-COR 926–68070, 1:5000), IRDye 800CW Goat anti-Rabbit IgG (LI-COR 926–32211, 1:5000). Fluorescent signals were detected using a LI-COR Odyssey CLx imager.

### Phenotypic responses to genotoxic stress

Fungal cells were grown on YMS plates for three to four days and diluted to $OD_{590} = 1$ in water (~$10^7$ cells/mL and tenfold dilution series to 100 cells/mL); 3 μL of the spore dilutions were pipetted onto plates. Pictures were taken after 7 and 12 days of growth (18˚C, 28˚C, and RT [~23˚C]). To test for responses to different genotoxic stressors *in vitro*, YMS plates containing hydroxyurea (HU; 100 μM, 1 mM, 10 mM), camptothecin (1 mM, 10 mM, 100 mM), and methyl methanesulfonate (MMS; 0.005% and 0.0075%) and three plates containing only YMS were prepared.

### Data analyses

Detailed descriptions of software and commands for analyses of ChIP-seq and RNA-seq data are included as S5 Method and S6 Method. A list of all sequencing datasets can be found in S5 Table.

### ChIP-seq

ChIP-seq data were quality-filtered and adapters removed with trimmomatic v.0.39 [69]. Mapping was performed with bowtie2 v.2.4.4 [70]. Sorting and indexing was performed with samtools v.1.9 [71]. Normalized coverage bigwig files and heatmaps were created with deeptools v.3.5.1 [72]. ChIP-seq data was normalized by RPGC (1x coverage). Wiggletools v.1.2 and the UCSC Genome Browser tools were used to calculate means for replicates and converting wig to bigwig files. Peaks were called with homer v4.11.1 [73] and bedtools v2.30.0 was used to make windows, calculate genome coverage and intersect peaks with genomic features and to identify *ash1-* and *set2*-specific H3K36me3.

### RNA-seq

RNA-seq data were quality filtered and adapters removed with trimmomatic v.0.39 [69]. Mapping of all paired reads was performed with hisat2 2.1.0 [74]. Sorting and indexing was performed with samtools v.1.9 [71]. HTSeq v.0.11.2 [75] was used to count reads on features. The *Z. tritici* IPO323 gene annotation was obtained from FungiDB (release 53) [76], based on previous annotations [77]. Differential expression analyses were conducted with DEseq2 v.1.32.0 [78]. New transcripts were annotated by combining all mapped RNA-seq data with stringtie v. v2.2.1 [45]. Manual annotation was done by combining all RNA-seq data and extracting transcript sequences that were > 200 bp in length and supported by at least 10 reads with IGV [79] and Geneious [80]. Transcripts were characterized by blastn with the *Z. tritici* genome as a reference to find homologous sequences on core and accessory chromosomes and blastx to find homologs in other species and to infer functional annotations.

### Supporting information

**S1 Table. Genome-wide and per chromosome bp coverage of H4K20me3, H3K36me3 and H3K27me3 in wild type, Δ*kmt5* and Δ*ash1* strains.**
(XLSX)

**S2 Table. DESeq2 results for all mutants in comparison to wild type including functional annotation.**
(XLSX)

**S3 Table. Detailed overview of all manually annotated transcripts on chromosome 17.**
(XLSX)

**S4 Table. List of all plasmids, strains, and primers generated and used in this study.**
(XLSX)

**S5 Table. Summary of all sequencing datasets generated in this study.**
(XLSX)

**S1 Method.** *Zymoseptoria* **transformation protocol.**
(PDF)

**S2 Method. ChIP protocol.**
(PDF)

**S3 Method. Modified NEB Illumina library protocol.**
(PDF)

**S4 Method. Stranded mRNA library protocol.**
(PDF)

**S5 Method. ChIP-seq analysis.**
(TXT)

**S6 Method. RNA-seq analysis.**
(TXT)

**S1 Fig. Comparison of H4K20 HMT SET domains.** A) Alignment of SET domains of H4K20 methyltransferases in different species. SET domain of human EZH2 is used as an outgroup. B) Phylogenetic tree based on the alignment in A. Numbers indicate substitutions per site. C) Accession numbers of proteins used for analyses shown in A and B.
(PDF)

**S2 Fig. Uncropped western blots for detection of H4K20me1, H4K20me3, and H3.**
(PDF)

**S3 Fig. Southern blots to confirm deletion of** *kmt5* **and** *ash1*. A) Δ*kmt5* mutants show more bands in addition to the expected bands. Because one of the flanks used for homologous recombination and as a probe for Southern analyses is a repetitive element, integration into the repetitive element may have altered restriction sites, and additional bands are detected because of TE copy number. Deletion of *kmt5* was confirmed in all relevant ChIP- and RNA-seq datasets. Strains used for further analyses were 95 and 100. B) Confirmation of Δ*ash1*. Strains used for further analyses were 5 and 25. Deletion of *ash1* was confirmed in all relevant ChIP- and RNA-seq datasets.
(PDF)

**S4 Fig. Phenotypic characterization of wild type (WT), Δ*kmt5*, Δ*ash1*, K20M, K20/K20M, and Δ*eaf3* under different genotoxic stress conditions and temperatures.**
(PDF)

**S5 Fig. Analysis of** *kmt5* **and** *ash1* **complementation strains.** A) ChIP-seq shows wild type-like enrichment of H4K20me3 and H3K36me3 for both, *kmt5* and *ash1* complementation

strains. The *ash1* complementation strain is lacking chromosome 16. B) Phenotypic assays performed at 18˚C and RT (23˚C) testing genotoxic stresses. YMS plates were used as a control. Pictures were taken seven days after inoculation. Complementation strains exhibit wild-type growth.
(PDF)

**S6 Fig.** Distribution of H4K20me3, H3K36me3, H3K27me3, H3K9me3, and H3K4me2 in wild type on chromosome 2 as an example region A) and genome wide B).
(PDF)

**S7 Fig. Enrichment of different histone marks outside A) and within B) facultative hetero-chromatin regions in all deletion mutants.** While we observed subtle changes outside of facultative heterochromatin, we observed the most prevalent differences in regions of facultative heterochromatin.
(PDF)

**S8 Fig. Genome-wide distribution of A) H4K20me3, B) H3K27me3, and C) H3K36me3 in all mutants generated in this study.** There are no ChIP-seq data for H3K36me3 in the tagged Kmt5 strains. [1]tagged *kmt5* was integrated in the Δ*kmt5* strain; [2]tagged *kmt5* was integrated in the wild-type strain.
(PDF)

**S9 Fig. Phenotype of Δ*set2* mutants.** A) The Δ*set2* mutants grow very slowly. Growth appeared ~ 14 days post inoculation on YMS and incubation at 18˚C (compared to ~2 days for wild type) and was limited to sparse hyphal growth, as shown in B) by microscopy. In contrast, the wild type predominantly produces spores under these conditions; Δ*set2* mutant seemed unable to produce spores.
(PDF)

**S10 Fig. Percent coverage of bp of histone marks H3K27me3, H3K36me3 and H4K20me3 in wild type, Δ*kmt5* and Δ*ash1* strains. Peaks were called using HOMER [73] and genome coverage of enriched regions was calculated with bedtools genomecov [81].**
(PDF)

**S11 Fig. ChIP-seq of the Δ*pdp1* mutant revealed an increase in H3K27me3 outside of facultative heterochromatin regions in wild type (WT).** A) ChIP-seq tracks (chromosome 10 as an example region) show that H3K27me3 enrichment increases in gene-dense regions that are also enriched with H4K20me3 and H3K36me3 but not H3K9me3.
(PDF)

**S12 Fig. RNA sequencing of mutant strains.** A) Principal Component Analysis (PCA) plot of all sequenced replicate strains. B) Venn diagram of upregulated genes (DESeq2, padj $\leq$0.01, | log2 fold change | $\geq$0.585) in all mutants compared to wild type (WT).
(PDF)

**S13 Fig. Alignment of sequences showing similarity (red) to the gene encoding Shugoshin on core chromosome 12 that were identified in newly annotated transcripts (purple) from chromosome 17 to the *shugoshin* gene sequence.** Pairwise sequence similarity is noted below the alignments and ranges from 55 to 78%. Light grey represents agreements to the reference and black represents disagreements.
(PDF)

## Acknowledgments

We thank Eva Stukenbrock for supplying materials and strains, and for supporting this work while MM was still at CAU Kiel. We thank colleagues in the Freitag lab and Zachary Lewis (University of Georgia) for conversations and comments on the manuscript.

## Author Contributions

**Conceptualization:** Mareike Möller, Michael Freitag.

**Formal analysis:** Mareike Möller, Michael Freitag.

**Funding acquisition:** Mareike Möller, Michael Freitag.

**Investigation:** Mareike Möller, Devin F. Wright, Faith A. Martin, Michael Freitag.

**Project administration:** Michael Freitag.

**Resources:** Mareike Möller, John B. Ridenour.

**Supervision:** Michael Freitag.

**Visualization:** Mareike Möller.

**Writing – original draft:** Mareike Möller, Michael Freitag.

**Writing – review & editing:** Mareike Möller, John B. Ridenour, Devin F. Wright, Faith A. Martin, Michael Freitag.

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
