## [Decision Letter · Decision Letter 0]

25 May 2023

Dear Dr Möller,

Thank you very much for submitting your Research Article entitled 'H4K20me3 controls Ash1-mediated H3K36me3 and transcriptional silencing in facultative heterochromatin' to PLOS Genetics.

The manuscript was fully evaluated at the editorial level and by independent peer reviewers. The reviewers appreciated the attention to an important topic but identified some concerns that we ask you address in a revised manuscript.

We therefore ask you to modify the manuscript according to the review recommendations. Your revisions should address the specific points made by each reviewer. 

I believe it is particularly crucial to: 1. Include the additional clarifying information suggested by Reviewer #1 regarding the role of non-coding RNAs and the changes in H3K27me3 levels in the absence of Kmt5. 2. Assess and discuss whether Kmt5 or Ash1 complementation strains are capable of restoring wild-type phenotypes.

Yours sincerely,

Alessia Buscaino

Guest Editor

PLOS Genetics

Geraldine Butler

Section Editor

PLOS Genetics

Reviewer's Responses to Questions

**Comments to the Authors:**

Reviewer #1: Review of “H4K20me3 controls Ash1-mediated H3K36me3 and transcriptional silencing in facultative heterochromatin”

Möller et al.

PLoS Genetics

050823

In the manuscript “H4K20me3 controls Ash1-mediated H3K36me3 and transcriptional silencing in facultative heterochromatin”, Möller et al. characterize the histone post-translational modification (PTM) H4K20me3 as being important for the establishment and transcriptional repression of facultative heterochromatin in the filamentous fungal organism Zymoseptoria tritici, a critical pathogen of wheat. The authors show that the deposition of H4K20me3 by the enzyme KMT5 controls the deposition of H3K36me3 by ASH1 and can influence the enrichment of H3K27me3. Together, these data argue that the action of KMT5 to trimethylated H4K20 is epistatic to ASH1 deposition on H3K36 and can influence the activity of the PRC2 complex for trimethylation of H3K27; these conclusions were reached using both strains containing full deletions of the histone methyltransferases and in strains encoding histones with the requisite lysine mutated (either as the only copy or as ectopic copies in tandem with the endogenous histone H4 gene). The loss of H4K20me3 and H3K36me3 in Δkmt5 strains and movement of H3K27me3 then influence the expression of genes in formerly facultative heterochromatic regions, with hundreds of non-coding genes (based on the presence of stop codons in their reading frames) that had homology to protein-encoding genes on the core chromosomes being upregulated. These novel transcripts argue that Z. tritici employs transcriptional gene silencing with these non-coding transcripts to modulate gene expression, possibly in different developmental stages or in response to different environmental conditions - a fascinating mechanism that should be explored further in later manuscripts.

Overall, I felt this manuscript presented a clear story of the histone PTMs (“epigenetics”) of Z. tritici and the epistatic nature of the H4K20me3 mark controlling ASH1-specific H3K36me3 and influencing H3K27me2/3 deposition and gene expression. In general, the authors performed multiple ChIP-seq and RNA-seq experiments that were convincing of the role of these enzymes in establishing the repressive facultative heterochromatin. The authors also added in Hi-C data in the supplemental data, and given the paucity of information about it, it seems appropriate to have these data in the supplement (although I have questions about it). The genomics experiments are sufficient for this manuscript, and while I think some biochemical experiments could enhance the story, I agree with the authors that they are better suited for a later manuscript. Overall, I think PLoS Genetics is the appropriate journal for this manuscript.

However, I feel there are some places in this manuscript where the authors could have elaborated with additional clarifying information. Also, another figure that specifically shows the potential interactions between their novel non-coding RNAs and the corresponding protein-encoding genes in the core chromosomes would go a long way towards explaining this observation better. Specifically, more information is needed about the non-coding RNAs in facultative heterochromatin, including directly stating that these are polyadenylated, given your methods to isolate these non-coding RNAs use oligo dT. Why are they polyadenylated (perhaps the authors could speculate in the discussion)? Would these non-coding RNAs be thought to act through basepairing interactions with their targets? Would they be expected to be processed into small interfering RNAs by Dicer or Argonaut (or, even more basic, are there homologs for Dicer and Argonaut encoded in the Z. tritici genome)? This specific section must be elaborated with additional text/information so that it is abundantly clear to the reader how these non-coding RNAs were discovered (without having to dig deep in the supplement to find out that they are polyadenylated) and what these non-coding RNAs could be doing. Some information that details the action of those non-coding RNAs would be beneficial to discuss beyond the one sentence that was currently written in the Discussion ([lines 472-474]: “How exactly these RNA species act, what their specific functions may be, and whether they interfere with or aid propagation of accessory chromosomes in Z. tritici are interesting questions but go beyond the scope of the current study.”).

I did have some major and minor concerns regarding this manuscript, which if addressed, should not impede publication:

Major concerns:

1. As detailed above, my biggest concern came with the paucity of information regarding the non-coding RNAs encoded on the accessory chromosomes in Z. tritici. The paragraph (lines 320-348) describing the identification of these non-coding RNAs really piqued my interest, but there is no evidence associated with it. I think that some (two?) alignments of non-coding RNAs (with stop codons boxed) and their possible protein-encoding binding partner on the core chromosome could be shown to elaborate on these observations (perhaps that is what Figure 6C is for; this figure is not referred to in the manuscript at all). The Results section text should also specifically state that these non-coding transcripts must be polyadenylated given the protocols used for isolating and sequencing these RNA species, as well as how they were discovered: if these transcripts are homologous to genes on the core chromosome, how did they map to the accessory chromosome? I am assuming there was enough sequence divergence to allow mapping to two different sites in the Z. tritici genome? This needs to be fully explained (again, perhaps Figure 6C is meant to elaborate this point, but if so, the same read mapping data should be shown for the core chromosomes to highlight the unique reads present at those two loci - and zoom in so that two or three genes are shown). And as noted above, some speculation about the mechanism of processing and targeting for these non-coding, regulatory transcripts would be beneficial to the reader. As presented, this section is insufficient for an understanding of this observation.

2. As alluded to above, the Hi-C data does not have enough information for interpretation. As presented, you have an observed vs. expected contact map, which shows some facultative heterochromatic regions having less interactions with surrounding chromatin. However, additional information is needed for readers to interpret these data, including the number of valid reads produced from the experiment that went into the contact matrix (not just the toal number of reads) and the resolution of the images (what are the bin sizes?), as well as images of the contacts across the genome in raw and Knight-Ruiz corrected datasets (are the matrices used for the observed-expected heatmap presented in the supplemental information Knight-Ruiz corrected? This is critical to eliminate biases in the contact heatmap that might be due to underlying sequence differences or differences in DpnII site locations), which would be critical to understand if the Z. tritici genome also employs a Rabl chromosome conformation like other fungal systems (e.g., S. pombe, S. cerevisiae, N. crassa, V. dahliae, etc.); the heatmaps shown here look like only a couple of chromosomes have contacting centromeres, which would be unique among other high-quality fungal Hi-C datasets where all centromere contacts are uniform between all chromosomes. The authors also need to state in their manuscript, preferably in the results section, that the Hi-C data was collected using proximity ligation, which is known to be less accurate than in situ Hi-C in which ligation products are captured in the nucleus and as such are more apt to reflect valid contacts. This information would be helpful to clarify the genome organization of Z. tritici, which to my knowledge, is not known.

Lastly, lines 173-176, in the authors state that the facultative het region interactions persist in the absence of H3K27me3, but from what I can tell, the Hi-C data presented are just of a WT strain. How can the authors conclude that H3K27me3 plays no role? Am I missing a manuscript in which this is referred (if so, please add that information)? Please revise this entire section to more accurately detail the Hi-C data, and/or to explicitly detail the Δkmt6 Hi-C data if it is presented here.

3. Do the authors have any idea why H3K27me3 is not ablated in a Δkmt5 mutant? Certainly, with the loss of H4K20me3, some regions lose H3K27me3 and others have a slight redistribution of that mark. These data argue that there is some (probably indirect) control of H3K27me3 by KMT5, but it is unclear to what extent. Could the authors speculate in the discussion more about what the redistribution of H3K27me3 could mean? Is it possible that the PRC2 complex binds H4K20me3 for normal localization? Is there information about whether PRC2 interacts with Eaf3 or Ash1, possibly from other species? At the very least, there should be a greater discussion of the results in Bicocca et al., which detail how some facultative heterochromatic regions in N. crassa have co-localization of both ASH-1 specific H3K36me2 and H3K27me2/3; it is possible that the facultative heterochromatin regions most affected in a Δkmt5 strain could be those that present with overlapping H3K36me3 and H3K27me2/3. Perhaps the authors could take a bed file with H3K36me3 domains and ask whether the H3K27me3 enrichment is strongly changed in these regions in a Δkmt5 strain relative to a WT strain? In general, the changes of H3K27me3 in a Δkmt5 strain should be addressed a more in this manuscript.

Minor concerns:

1. Figure 6C is not referred to in the results text. Please add that text, in addition to the suggestions I have provided above.

2. I think the manuscript would be better off with a model of possible action. You have one in Figure S11; I don’t think this should be buried in the supplemental material. I would suggest moving that figure to the main figures. Doing so will help the reader understand the epistatic depositions of histone PTMs in facultative heterochromatin.

3. Line 41: “…to understand epistasis relationships…” epistatic?

4. Line 46ff: this run-on sentence is really long and quite difficult to understand. Please break it up.

5. Line 233ff: is there any information on what these regions could be doing to still have H4K20me3 in the cells with two copies of the histone H4 genes? Are there specific signals? Correlated to epigenetic marks or Hi-C contacts? Perhaps add any information you have to the discussion.

6. Line 235: the authors state that the “…presence of H4K20me3 appears to be necessary to deposit H3K36me3 and in some regions H3K27me3”. But there are still extensive regions of the genome with enrichment of the downstream regions. As stated, this seems like an “all or none” effect, but it’s more complex than that. Please clarify and/or revise this statement.

7. Line 243ff: please clarify with fusion (N or C or both?) of KMT5 produces that observation noted in this sentence.

8. Line 294ff: the authors denote the number of genes with changes in gene expression in a Δkmt5 strain relative to a WT strain. Are all these genes found in facultative het regions, or are they enriched with H4K20me3, H3K36me3, or H3K27me3? It is unclear if these numbers are the total changed genes or just those in facultative het regions (as written, I would believe the former). It would be great for the authors to clarify how many genes may have direct changes in expression, specifically reporting the genes that are enriched with any (or all) of the facultative heterochromatin marks and showing an example or two of genes that could be subject to direct changes in facultative heterochromatin.

9. Line 425ff: as written, this statement suggests that Eaf3 somehow represses KMT5 activity. Is this what the authors are concluding (e.g., without Eaf3, KMT5 is capable of continued trimethylation)? Please clarify.

10. Line 497: please subscript the “2” in MgCl2.

11. The reviewer links for the NIH Bioproject do not work (they seem to be duplicate links), but I can access the zenodo data. What I really wanted to see was whether the authors provided the .bed files for the genome features they evaluate here, as well as the contact matrices for your Hi-C data, which are not present in the zenodo database (but may be in the SRA [the GEO might be better?], perhaps as supplemental files somehow?). Please make sure you provide not just the sequencing reads, but the Hi-C contact matrices and the bed files for enriched regions to your readers.

12. In figure 4A, the authors have ChIP-seq tracks with different y-axis scales. Perhaps it would be best to set all scales to the same range for accuracy – this would be important to show how the K20L mutation might seem to cause increased enrichment of H3K20me3, possibly suggesting a stronger interaction of KMT5 with these regions (maybe these data speak to the binding specificity of KMT5?)? Lastly, please be sure to explicitly detail, perhaps in the Materials & Methods, how your ChIP-seq data was normalized (by RPGC, per your supplemental data), so readers know what normalization method was applied.

Reviewer #2: The manuscript ‘H4K20me3 controls Ash1-mediated H3K36me3 and transcriptional silencing in facultative heterochromatin’ by Möller and colleagues provides evidence that the histone modifications H4K20me, catalyzed by Kmt5, and H3K36me3, catalyzed by Ash1, are important for facultative heterochromatin in the fungal plant pathogen Zymoseptoria tritici. The manuscript reports on a combination of genomic, genetic, and biochemical experiments to show that the deletion of kmt5 leads to de-repression of expression, which was accompanied by loss of H3K36me3, suggesting that H4K20me3 is important for Ash1-mediated H3K36me3. Loss of H3K36me3 and H4K20me3 led to chromatin reorganization at facultative heterochromatic regions, defined by H3K27me3, H4K20me3 and H3K36me3 in the wild-type strain, thereby providing a mechanism for formation and maintaining of heterochromatin in filamentous fungi.

While the roles of H3K27me3 H3K36me3 in several fungi has been reasonably-well established, the function of H4K20me3 has thus far received far less attention, and the data presented here will be of interest for researchers working on fungal chromatin biology. This submission is generally well-written, the data seems to be robust, and the applied approaches are generally well-suited to support the manuscript’s results and conclusions. I would also like to applaud the authors to not only provide the sequencing data but also the scripts as part of the submission.

I, however, also have few essential comments that necessitates the authors’ attention:

Loss of Kmt5 or Ash1 lead to several measured phenotypes such as loss of specific histone modifications, changes in chromosomal stability or gene expression changes. Throughout the manuscript, no Kmt5 or Ash1 complementation strains were systematically tested for their capacity to restore wild-type phenotypes. It is mentioned that complementation with tagged Kmt5 does not restore wild-type levels of H4K20me3 (L238fF), which could either indicate that the tags interfere with the correct function or that additional, ectopic effects in occur in the mutants independent of Kmt5. To address this possibility, and I consider complementation experiments essential.

Based on the description in the manuscript, it seems that the ChIP-seq analyzes were not performed with controls. Since it is not entirely clear if the data has been corrected by e.g., input or no-antibody control, some of the visual enrichments for specific modifications might also be due to background effects. Related to this, I am a bit puzzled why H4K20me3 displays clear enrichment in specific regions, while H4K20me1 does not seem to be enriched (L145). Is it possible that H4K20me1 is largely absent, and that the genome-wide data shows unspecific background? Figure 1 obviously shows the presence of H4K20me1 but are the antibodies specific enough to distinguish H4K20me1 and me3?

The manuscript compares protein sequences between different species and inferences potentially diverged functions based on the absence of conserved protein domains or motifs (e.g., L114). Simple sequence similarity searches can easily miss domains/motifs and structural comparisons (e.g., using structure predictions from Alphafold) are much more sensitive to identify diverged yet structurally conserved motifs/domains. To corroborate that absence of domains is linked to divergent functions, structural comparisons of the enzymes studied here would be essential.

The manuscript concludes ‘A homolog of Z. tritici Pdp1 was found to interact with a chromatin remodeling factor, ISWI, in N. crassa, where it also increased H3K27me3 levels; thus Z. tritici Pdp1 is a putative homolog of S. cerevisiae Ioc4.’ Based on the information provided, it is not clear how this conclusion is derived? Is loc4 a homolog/ortholog of Pdp1 in Neurospora/Z. tritici? It seems that the results and discussion here focus on functional equivalence rather than evolutionary relatedness, the latter which is key to determine if sequences are homologs/orthologs. Thus, what is a ‘true ortholog’ or ‘best homolog’ (L432) according to the authors? To disentangle this in the results and discussion, the manuscript needs to show an alignment of PWWP domain containing proteins (or just the domains) and a phylogenetic analysis to determine which sequences are orthologs, and then superimpose the potential function; proteins with distinct functions can be orthologs and homologs with similar function might still be paralogs.

The importance of the co-localization of heterochromatic regions to the presented story is not clear. Co-localization of H3K27me3-rich regions have been previously reported for some fungi (e.g., Neurospora crassa or Verticillium dahliae), and thus the observation that regions enriched for H3K27me3, and with H3K36me3 and H4K20me3 as a consequence of the overlaps, are interacting is not surprising per se, as is the notion that interactions are maintained in the absence of H3K27me3 (has been shown previously for N. crassa). However, Kmt5 and/or Ash1 mutants were not assayed with Hi-C, and thus their role in maintaining the 3D chromatin organization remain unclear. These experiments would be essential for the presented narrative, and in the absence of these, I would suggest removing these data from the current submission.

Lastly, I would suggest adjusting the manuscript’s title such that it clarifies that the manuscripts focused on filamentous fungi.

Additional detailed comments and suggestions aimed to further improve the manuscript (in no specific order):

L19: ‘Reversal of silencing…’ would it maybe be clearer to say ‘De-repression…’ ?

L49: ‘…specific genes…’ What is meant here by specific genes? Could the manuscript please be more precise what defined these set of genes?

L76-82: it should be clarified that most fungi, if not all, have SET2, but that only a subset seem to also have ASH-1 (see e.g., Bicocca et al., 2018, Elife; Zhang et al., 2022, Microb. Genomics). The emphasis here is on ASH-1, which makes sense given the data presented, but the current narrative might suggest that species might only have ASH-1, which should be clarified. Moreover, the importance and function of SET2 and the interaction between H3K4me and SET2-catalyzed H3K36me should be explained more extensively. Lastly, the distinct functions of SET2 and ASH-1 should be mentioned (see e.g., Ren, et al., 2021, Journal of Fungi; Ferraro, et al., 2021, BMC Genomics).

L109: the manuscript cannot claim that Kmt5 is reasonable for all H4K20 methylation (Figure 1) as H4K20me2 was not assessed. To support this statement, additional ChIP-seq experiments and/or (at least) Western blots (as in Figure 1B) are required; alternatively, statements concerning the breadth of activity of Kmt5 need to be toned down.

L128ff: Loss of Kmt5 and Ash1 increases chromosome loss compared with the wild-type situations (Table 1). Can the authors speculate why specific chromosomes display increased loss rates? What are their features that distinguishes them from other accessory chromosomes.

L140: it would increase readability if the manuscript would directly explain which histone modification were assayed using ChIP-seq.

L158: The co-occurrence of H3K9me3 and H3K27me3 at specific loci has been previous reported by Schotanus and colleagues (2015), which should be also mentioned here.

L157: The composition of cluster 1 and 2, also in respect to the precise numbers/fractions of other histone modifications such as H3K9me3 need to be reported.

L190: the manuscript indicates that '…at least one H3K36 methyltransferase is essential.' Based on the phenotypes of the mutants, it seems that indeed SET2 and not ASH-1 is essential, which is in line with previous work (see above), and should be clarified.

L217: why was lysine 20 replaced by these specific amino acids. Could the manuscript please provide some insights?

L293: it is not clear if the 718 reported genes are differentially regulated or indeed up regulated in the mutant. The text says the former, the notion that the log2FC > | 0.585 | suggests that latter. Please clarify. Where is the cutoff (0.585) coming from?

L323: Why focus on chromosome 17 for manual annotation and not for the other accessory chromosomes? Would the picture look any different if more or different accessory chromosomes would be analyzed? What is the function of the anti-sense transcripts if no correlation with gene expression levels of homologs genes could be observed? This section necessitates either a more systematic analyzes or should be significantly shortened.

L416: I would suggest, if possible, to move the model in Figure S11 to a main figure to further support the clarity of the discussion.

Figure 1: Does Neurospora crassa lack KMT5? Otherwise, it would be useful to add the cartoon of the protein here as well. Moreover, it might be useful to swap panel B and C to fit the overall narrative of the manuscript.

Figure 2: I would suggest changing the order in panel to H3K27me3, H3K27me2, H4K20me3, H4K20me1, H3K36me3, H3K9me3, and H3K4me2 to better fit the narrative in the manuscript.

Figure 3: Transposable elements should be shown, analogous to the display in Figure 2. It is not clear how the fragments in the heatmap in B was ordered; I assume from high to low for the WT, yet this is not fully explained (this also applies to the other figures).

Figure 4: in panel C and D it is unclear which of the heatmaps display the wild type. Could these please be added (either via labelling or by adding the wild-type data for clarity)?

Figure 5B: The wild-type heatmaps seems to differ from the one in Figure 3. How is this possible as supposedly the same data is shown?

Figure 6: from panel B it is not entirely clear if all genes are coming from H3K27me3 enriched regions or only a subset of those shown. Could the numbers be added to the legend or figure?

Reviewer #3: The manuscript by Moller and colleagues reports on genetics, genome-wide distribution, transcriptional impact, and biological aspects of heterochromatin formation in the fungal pathogen Zymoseptoria tritici. The focus of the research is the interplay between H4K20me3 and H3K36me3 deposited by Kmt5 and Ash1 respectively. Understanding heterochromatin formation and function is of great importance to genome biology, and of specific interest in fungal biology. There are many questions related to fungal development and evolution, along with environmentally regulated transcription. The results reported here provide new insights into hierarchy and cross-talk between histone modifications contributing to heterochromatin formation. I enjoyed reading the paper, and it makes important contributions to the literature of heterochromatin function.

I read the included reviews from previous submission, which did raise valid concerns. However, I agree with the authors assessment that they are good suggestions for future work. The genetic conclusions regarding ‘requirement’ of specific proteins for individual histone modifications can stand on its own, provided the conclusions are discussed as such.

My overall comments regarding the manuscript are related to the title and reliance on chromatin maps to make genome-wide claims about requirements, interactions, and functional impacts. The word ‘control’ in the title is too strong given the previous reviewers concerns, and some contradicting data in the paper. The results and claims would be more interpretable with clearer quantification of chromatin domains, their changes in mutants, and reported impacts. Specific comments are listed below.

1. Lines 96-98: “Kmt5 and H4K20me3 are crucial for transcriptional silencing in regions of facultative heterochromatin and essential for Ash1-mediated H3K36me3, thereby contributing significantly to formation of H3K27me3-enriched facultative heterochromatin.” I am not sure why the authors use the term “H3K27me3-enriched facultative heterochromatin”. Is the point to link the results directly to H3K27me3, or to try and distinguish this specific type of heterochromatin? The point about different types of heterochromatin is a bit of a stumbling block for me in the paper. It is clear that not all heterochromatin is the same, but are there actually good data in the literature regarding ‘facultative’ heterochromatin in fungi? I interpret the term to imply dynamics, where the chromatin is switching states, but this lack empirical data in fungi. Or the authors can provide references. Otherwise, it seems important to note that the description of facultative heterochromatin is not actually about labile chromatin or DNA accessibility, but rather the presence of histone marks.

2. It is not a major conclusion in the paper, but it is not possible to assess the comments regarding Hi-C and interdomain interactions (Lines 171-175). There needs to be some analysis to support the statement about strong interactions between cluster 1 regions. It might also stand better to leave the impact of HiC for a different paper with more complete analysis on all the types of interactions. Interpretation of this type of figure is too subjective.

3. Have the authors considered calling domain peaks, such as with the common and well documented tool MACS2? This is not without its caveats, but the reader needs some quantitative/qualitative assessment of changes between these mutants to assess the genome wide claims made.

4. Can the authors provide some assessment of how much H3K36me3 signal is lost in ash1-KO. I appreciate they do not think WB will be informative, but the genome wide ChIP can be used to directly assess this. Either base pairs covered in WT versus ash1, or the number of peaks lost? This would help confirm that Ash1, and presumably Set2, function similar to reports in other fungi in terms of genome coverage and chromatin compartments.

5. Following from the previous comment, the analysis would benefit from clearly distinguishing H3K36me3 that is Ash1-dependent versus not dependent, and how this relates to chromatin and H4K20me3. For instance, in Fig. 3A, the left panel shows that the majority of H3K36me3 in the region is Ash1-dependent, kmt5-dependent and overlaps H4K20me3. But in the right panel, it is not so clear. Part of the WT H3K36me3 appears Ash1 and Kmt5-dependent, although a section is clearly not, while the whole section contain H4K20me3. I cannot find anywhere in the paper how often H3K36me3 and H4K20me3 overlap, how this relates to the chromatin designation, and how it relates to the H3K36me3-dependence on H4K20me3. Does H4K20me3 occur with non-Ash1 H3K36me3 when it is outside of defined facultative heterochromatin? This is directly relevant, for instance, to the discussion 388-392. In other fungi, Ash1-H3K36me3 is not associated with transcription.

6. Related to the previous suggestion, kmt5 and ash1 KOs have similar numbers of DEG, but the overlap of regulated transcripts is ~<50%. Do the authors think this point is reflected in the title of the paper- ‘H3K20me3 controls Ash1 and transcriptional silencing’. There is not a direct analysis for ash1KO DEG that reports how many of these genes overlaps H4K20me3 and reside in H3K36me3 regions dependent on kmt5. If the other 50% of ash1 KO transcripts are independent of H4K20me3/Kmt5 it seems that point needs to be made more clear.

Minor:

1. Line 135-136 “[…] and more likely general genome stability.” Why would this data point to general genome stability? The effects are quite specific to specific chromosomes of the accessory chromosomes checked. Maybe the statement is true, but I don’t see how the presented data leads to this as a logical conclusion.

2. It is hard to know what the significance is of the increased transcription and newly annotated genes on the accessory in kmt5KO. It is interesting, I support its inclusion in the paper, but the text could be much shorter. The observations are interesting, but incomplete, and may contribute to the previous reviewer negative comments. There is very little text in the discussion regarding these findings. I suggest shortening the results text, lines 280-348.

Reviewer #4: This manuscript focuses on the role of H4K20me3 and Ash1-mediated H3K36me3 in facultative heterochromatin in the important wheat pathogen Zymoseptoria tritici. The authors identify the lysine methyltransfersases (KMTases) responsible for H4K20me- and heterochromatin-associated H3K36me (Kmt5 and Ash1 respectively) and show that loss of these KMTases results in increased loss of accessory chromosomes. Using ChIP-seq they map the genome-wide distribution of H4K20me3 and show that Kmt5 is required for the proper distribution of H3K27me3 and Ash1 mediated H3K36me. Importantly, H4K20me is required for silencing and deletion of kmt5 results in the de-repression of genes located in facultative heterochromatin. They also demonstrate that the Z. tritici Eaf3/MRG15 homologue is required for Ash1 activity at H4K20me-enriched regions. Overall, this is a high-quality study that is important for several reasons. The function of H4K20me in fungi is very poorly understood and as the authors point out, genome-wide analyses of H4K20me distribution have not been performed even in model fungal systems. Furthermore, there is little known about the interplay between H4K20me3 and H3K27me marks and also this study explains why the loss of H3K23me3 alone has only a minor effect on transcriptional activation from facultative heterochromatin in Z. tritici. I have a just few minor points that the authors may want to consider.

1) Were multiple isolates of the different (Δkmt5, Δash1, Δeaf1, K20M etc) mutants generated and examined? There can be variability in the phenotypes associated with different isolates. Although I am certainly not suggesting that experiments need to be repeated with multiple independent isolates, it would be good to know that an initial characterisation revealed consistent phenotypes.

2) The corresponding author has demonstrated that H3K27me3 promotes chromosome instability in Z. tritici. Do the authors anticipate that the increased accessory chromosome loss associated with Δkmt5 and Δash1 is dependent upon Kmt6/H3K27me3?

3) The authors report that loss of set2 leads to severe growth defects. It would be useful to have a supplementary figure of the Δset2 mutant phenotype.

4) What is the impact of deletion of kmt5, ash1, etc on virulence? Given the hypothesis that genes in facultative heterochromatin are differentially expressed during host colonisation (e.g. Miele et al. 2020 mBio), I am surprised that wheat infection assays with these mutants were not included.

5) Loss of kmt5 and ash1 results in novel transcripts from accessory chromosomes. Are these mutants also associated with an increased level of cryptic intragenic transcripts from bona fide genes on essential/core chromosomes?

6) Is it possible to show a composite plot of histone modifications for genes marked by H4K20me3, H3K27me3 and H3K36me?

7) Typo: Lines 456 and 457: italicise Δkmt5 and Δash1

**Have all data underlying the figures and results presented in the manuscript been provided?**

Reviewer #1: **No: **There are reviewer links for a Bioproject at the NIH SRA, but they do not work, and I cannot evaluate whether the data have been appropriately provided. Perhaps request to the authors a working link?

Reviewer #2: Yes

Reviewer #3: Yes

Reviewer #4: Yes

PLOS authors have the option to publish the peer review history of their article (what does this mean?). If published, this will include your full peer review and any attached files.

Reviewer #1: No

Reviewer #2: No

Reviewer #3: **Yes: **David E. Cook

Reviewer #4: No

---

## [Editor Report · Decision Letter 1]

30 Aug 2023

Dear Dr Möller,

We are pleased to inform you that your manuscript entitled "H4K20me3 is important for Ash1-mediated H3K36me3 and transcriptional silencing in facultative heterochromatin in a fungal pathogen" has been editorially accepted for publication in PLOS Genetics. Congratulations!

Yours sincerely,

Alessia Buscaino

Guest Editor

PLOS Genetics

Geraldine Butler

Section Editor

PLOS Genetics

Comments from the reviewers (if applicable):

**Data Deposition**

http://datadryad.org/submit?journalID=pgenetics&manu=PGENETICS-D-23-00458R1

**Press Queries**

---

## [Editor Report · Acceptance letter]

19 Sep 2023

PGENETICS-D-23-00458R1 

H4K20me3 is important for Ash1-mediated H3K36me3 and transcriptional silencing in facultative heterochromatin in a fungal pathogen 

Dear Dr Möller, 

We are pleased to inform you that your manuscript entitled "H4K20me3 is important for Ash1-mediated H3K36me3 and transcriptional silencing in facultative heterochromatin in a fungal pathogen" has been formally accepted for publication in PLOS Genetics! Your manuscript is now with our production department and you will be notified of the publication date in due course.

With kind regards,

Jazmin Toth

PLOS Genetics

On behalf of:
